# Explainable AI and optimized solar power generation forecasting model based on environmental conditions

**Rizk M. Rizk-Allah**[1,2,3], **Lobna M. Abouelmagd**[3,4], **Ashraf Darwish**[3,5], **Vaclav Snasel**[2], **Aboul Ella Hassanien**[6,7] *

1 Department of Basic Engineering Science, Faculty of Engineering, Menoufia University, Shebin El-Kom, Egypt, 2 Faculty of Electrical Engineering and Computer Science, VSB–Technical University of Ostrava, Ostrava, Czech Republic, 3 Scientific Research School of Egypt (SRSEG), Cairo, Egypt, 4 Misr Higher Institute for Commerce and Computers, Mansoura, Egypt, 5 Faculty of Science, Helwan University, Helwan, Egypt, 6 Faculty of Computers and AI, Cairo University, Giza, Egypt, 7 College of Business Administration (CBA), Kuwait University, Kuwait, Kuwait

* aboitcairo@gmail.com

**Data Availability Statement:** benchmark [36] https://www.kaggle.com/datasets/anikannal/solar-power-generation-data?select=Plant_1_Generation_Data.csv.

## Abstract

This paper proposes a model called X-LSTM-EO, which integrates explainable artificial intelligence (XAI), long short-term memory (LSTM), and equilibrium optimizer (EO) to reliably forecast solar power generation. The LSTM component forecasts power generation rates based on environmental conditions, while the EO component optimizes the LSTM model's hyper-parameters through training. The XAI-based Local Interpretable and Model-independent Explanation (LIME) is adapted to identify the critical factors that influence the accuracy of the power generation forecasts model in smart solar systems. The effectiveness of the proposed X-LSTM-EO model is evaluated through the use of five metrics; R-squared ($R^2$), root mean square error (RMSE), coefficient of variation (COV), mean absolute error (MAE), and efficiency coefficient (EC). The proposed model gains values 0.99, 0.46, 0.35, 0.229, and 0.95, for $R^2$, RMSE, COV, MAE, and EC respectively. The results of this paper improve the performance of the original model's conventional LSTM, where the improvement rate is; 148%, 21%, 27%, 20%, 134% for $R^2$, RMSE, COV, MAE, and EC respectively. The performance of LSTM is compared with other machine learning algorithm such as Decision tree (DT), Linear regression (LR) and Gradient Boosting. It was shown that the LSTM model worked better than DT and LR when the results were compared. Additionally, the PSO optimizer was employed instead of the EO optimizer to validate the outcomes, which further demonstrated the efficacy of the EO optimizer. The experimental results and simulations demonstrate that the proposed model can accurately estimate PV power generation in response to abrupt changes in power generation patterns. Moreover, the proposed model might assist in optimizing the operations of photovoltaic power units. The proposed model is implemented utilizing TensorFlow and Keras within the Google Collab environment.

**Funding:** The author(s) received no specific funding for this work.

**Competing interests:** The authors have declared that no competing interests exist.

**Abbreviations:** Definitions, Acronyms; AI, Artificial Intelligence; ANNs, Artificial Neural Networks; ARMA, Autoregressive Moving Average; LSTM, Long Short-Term Memory; BiLSTM, Bidirectional LSTM; BPNN, Back-propagation Neural Network; CNN, Convolutional Neural Network; COV, Coefficient of Variation; DL, Deep Learning; DT, Decision tree; EC, Efficiency Coefficient; EO, Equilibrium Optimizer; KNN, k-nearest neighbor; LIME, Local Interpretable and Model-independent Explanation; LR, Linear regression; MAE, Mean Absolute Error; MPPT, Maximum Power Point Tracking; PCC, Pearson Correlation Coefficient; PSO, Particle Swarm Optimization; PV, Photovoltaic; $R^2$, R-squared; RMSE, Root Mean Square Error; RNN, Recurrent Neural Network; SVM, Support Vector Machine; XAI, Explainable Artificial Intelligence; GWO, Grey Wolf Optimization; nRMSE, Normalized RMSE; nMAE, Normalized MAE; RE, Relative Error.

# 1. Introduction

The worldwide development of different energy resources and increasing energy demand due to industrialization and the growing global population have raised the world's need for electrical power generated [1]. Photovoltaic (PV) power units represent the mainstream of renewable energy technologies due to the characteristics of solar energy, such as being inexhaustible, clean, free-pollution, and environment-friendly. Therefore, high-tech countries worldwide have concentrated on spending on research and development while providing incentives to promote solar PV systems [2]. PV power unit entails the direct conversion of solar energy into electrical energy. When a semiconductor is exposed to sun radiation (n-and p-type silicon), electricity is produced as electrons flow between electrodes. Although the PV power plant is simpler to construct than a fossil fuel power plant, the PV power plant can be affected by the construction site, timing, size, and panel capability [3]. In addition, the electricity generated by the PV plant can fluctuate sporadically due to Unforeseeable and unmanageable meteorological factors which include solar radiation, temperature, humidity, wind speed, and cloud cover. Significant fluctuations in temperature and solar radiation can have a substantial effect on energy production [4]. Due to of the nature of these variables, PV power generation may become unstable with causing a reduction in PV output power or a sudden surplus. Moreover, this might lead to an imbalance between generating power and load demand, affecting the power grid's ability to operate and control [5]. If electricity generation is precisely forecasted, operation optimization techniques, like peak trimming and reducing the system's uncertainty for power generation, can be effectively adopted [6]. Therefore, a method for precisely forecasting the amount of produced energy is vital for industrial power system applications [7]. Precise forecasting is vital for improving the level of electricity delivered to a grid and reducing the costs associated with the general variability [8]. Additionally, it can be employed for a variety of operation and control tasks such as power scheduling in transmission and distribution grids [9].

Over the past few decades, researchers and engineers have been promoting the advantages of recent innovations in data science, machine learning, and artificial neural networks (ANNs) for predicting the power generated from photovoltaics. In this regard, the forecasting approaches can be categorized as physical methods, artificial intelligence-based methods, statistical methods, and ensemble methods [10]. Artificial intelligence (AI) approaches have the potential to be valuable tools for predicting solar power generation. This is because they can address the complex relationship between input and output data, which is nonlinear in nature. The primary techniques for short-term predictions include linear regression, autoregressive moving average (ARMA), support vector machine (SVM), time series modelling, and back-propagation neural network (BPNN), among others. Linear regression requires a substantial dataset, and the accuracy of the fitting results might be influenced by pathological data [11]. Auto-regressive integrated moving average (ARIMA) models rely only on past power outputs, which may lead to significant inaccuracies in predictions [12]. The SVM approach is not capable of efficiently handling huge volumes of data in terms of both training time and predicting accuracy [13]. Furthermore, the procedure of selecting the kernel functions is challenging due to its greater suitability for categorization [14]. In order to get a higher convergence rate, it is necessary to enhance the algorithm of a conventional Backpropagation Neural Network (BPNN) [15]. In addition, the Markov chain relies on a large dataset, yet it may still perform well even when there is missing data [16]. The solar power forecasting task has previously used the k-nearest neighbor (KNN) machine learning technique [17]. Boosting, bagging, and regression trees are other machine learning algorithms that have shown high accuracy and effectiveness.

The field of deep learning has gained significant attention due to its relevance in renewable power forecasting, specifically in wind power forecasts. However, it has been noticed that many ensemble models employed in previous studies do not incorporate deep learning (DL) techniques such as long short-term memory (LSTM) or gated recurrent unit (GRU) networks [18]. Moreover, Furthermore, these models may suffer from lower accuracy as a result of the limitations of traditional optimization techniques included into them to acquire the optimal internal parameters, such as being stuck in local minima and subsequently acquiring suboptimal parameters. Thus, this paper overcomes these issues by integrating the LSTM with the EO algorithm into the proposed model, which is then applied to accurately depict the relationship between solar output power and environmental factors.

Recently, there has been a growing interest in using deep learning models for data mining, regression, and feature extraction due to their capabilities [19]. The prevalent deep learning models utilized for predicting solar power generation comprise the deep neural network (DNN), Boltzmann machines, recurrent neural network (RNN), and deep belief network (DBN). RNN has emerged as the favored alternative for performing predictions in smart grids [20]. LSTM, a specialized form of RNN, has been utilized in research studies to enhance predicting accuracy when compared with standard ANN models [21]. Authors in [22] proposed a deep LSTM-RNN model for precise prediction of solar power output. While LSTM exhibits a significant level of predictive accuracy, it is characterized by a lengthy training duration. In [23], Authors suggested an integrated framework utilizing convolutional neural network (CNN) and bidirectional LSTM (BiLSTM) to precisely predict the energy output of a short-term photovoltaic system. After evaluating the model's accuracy, they concluded that the suggested CNN-BiLSTM model exhibits a much higher predictive influence compared to both the CNN and BiLSTM models. Nevertheless, the model is still subpar in terms of prediction accuracy, which may be caused by the smaller characteristics of the input data. Authors in [24] suggested a hybrid model using the full wavelet packet decomposition (FWPD) and the BiLSTM, named FWPD-BiLSTM, to estimate a day ahead solar irradiance. The FWPD-BiLSTM model has been demonstrated to be a highly effective forecasting model for improving the performance of solar irradiance predictions. Nevertheless, the optimization of hyper-parameters and the increased duration of execution represent significant hurdles in the implement of the suggested model. Study [25] examined eleven distinct forecasting models for point and interval forecasting of solar global horizontal irradiance (GHI) on an hourly basis, specifically for two locations in India. After investigating the model's accuracy, they observed that the BiLSTM model surpasses all individual models in terms of getting lower values for RMSE and MAE. Nevertheless, the study was challenged by the intricate hyper-parameter selection method and the significant amount of time required for execution. In [26], Authors offered a hybrid deep learning approach based on a robust local mean decomposition (RLMD) algorithm and the BiLSTM, named RLMD-BiLSTM, for accurate forecasting of solar GHI. The proposed hybrid model showed good accuracy in terms of RMSE and MAE over various contrast models, but hyper-parameter adjustment was selected by a grid search method, which is a time-consuming process. Moreover, it lacks the impact of combining other hyper-parameters like lag size, batch size, and drop period rate. Study [27] proposed a novel deep learning model for predicting solar power generation. The model includes data preprocessing, kernel principal component analysis, feature engineering, calculation, GRU model with time-of-day clustering, and error correction post processing. The findings of the experiments have shown that the suggested model exhibits superior forecasting accuracy compared to other conventional models and can produce outstanding prediction outcomes. Authors in [28] proposed a deep learning-based approach and a pre-processing algorithm to predict solar power. The reported results of the LSTM approach with adaptive moment estimation (ADAM) and root mean square

propagation (RMSP) show a good fit compared to other approaches. In [29], Authors provided a novel method for predicting global horizontal irradiance that is based on the LSTM and back propagation (BP), named LSTM-BP model, and the multi-physical process of atmospheric optics. The suggested model was compared to the LSTM model using comparable time scales and meteorological conditions. The suggested approach outperforms the LSTM model in clear, cloudy, and partly cloudy circumstances. The suggested approach improves prediction accuracy and expands its applicability. Authors in [30] presented a hybrid model based on deep learning techniques incorporating CNN and LSTM to forecast the short-term PV power generation at different times ahead. The proposed hybrid CNN-LSTM auto encoder approach surpasses the existing models in the literature in terms of the RMSE and MAE metrics. The suggested hybrid model achieves much lower values, varying from 40% to 80%, in comparison to other models documented in the literature. The extent of the reduction depends on the predicting interval. Authors in [31] suggested a LSTM model that is more effective at extracting temporal information compared to other deep learning models. This model is specifically designed to predict solar radiation data. The newly introduced model is referred to as the Read-first LSTM (RLSTM) model. The primary novelty of this study is the development of an enhanced LSTM model for predicting solar radiation data and the establishment of a collaborative procedure amongst gates. The provided findings indicate that the RLSTM model decreased the centralized RMSE of the BiLSTM, LSTM, RNN, and radial basis function neural network (RBFNN) models by 30%, 60%, 67%, and 70% correspondingly. The RLSTM, BiLSTM, LSTM, RNN, and RBFNN models had correlation coefficients of 0.99, 0.98, 0.96, 0.95, and 0.93, respectively. However, the RLSTM model necessitates the optimizer to tweak its hyper-parameters in order to further increase its accuracy. Authors in [32] proposed an innovative approach for predicting solar GHI 24 hours in advance by utilizing information from nearby geographic areas. The suggested methodology encompasses feature selection, data pre-processing, the utilization of Convolutional Long Short-term Memory (ConvLSTM) for feature extraction, and the implementation of a fully connected neural network regression model. The proposed method surpasses all other examined methods in terms of correlation coefficient and RMSE. Furthermore, the suggested model demonstrates superior performance compared to existing approaches. This confirms the effective attainment of the research objectives in forecasting solar GHI. However, the structural parameters of ConvLSTM lack optimization using evolutionary algorithms or other optimization techniques, which could enhance the accuracy of predictions. Authors in [33] introduced a hybrid model that incorporates an attention mechanism with the CNN and BiLSTM, named CNN-BiLSTM-Attention, for a short-term photovoltaic power prediction. This approach seeks to reduce the negative effects of weather variability on the precision of PV power prediction by efficiently extracting important characteristics from multidimensional time series data. The results confirm that the CNN-BiLSTM-Attention model provides outstanding performance compared to other models, but its performance is reliant upon a significant volume of training data, and the intricate nature of the model demands significant computational resources. Authors in [34] introduced a novel approach for PV power forecasting, combining federated learning (FL) and transfer learning (TL) in a hybrid deep learning model called Federated Transfer Learning Convolutional Neural Network with Stacked Gated Recurrent Unit (FL-TL-Conv-SGRU). This model addresses data privacy and security concerns while optimizing forecasting performance. Using a bio-inspired Orchard Algorithm (OA) for hyperparameter tuning and eight diverse PV datasets, the FL-TL-Conv-SGRU model trains in a federated manner, enhancing generalization and predictive capabilities. Empirical results show the model outperforms traditional methods, offering accurate forecasts and efficient, sustainable energy management while adhering to data protection regulations. Authors in [35] proposed a composite model for short-term wind

and PV power prediction, integrating LSTM and swarm intelligence algorithms to improve forecasting accuracy. This model leverages the Coati optimization algorithm (COA) to enhance hyperparameters of CNN-LSTM, leading to improved learning rates and performance. The results show a significant reduction in RMSE for day-ahead and hour-ahead predictions by 0.5% and 5.8%, respectively. The proposed COA-CNN-LSTM model outperforms existing models such as GWO-CNN-LSTM, LSTM, CNN, and PSO-CNN-LSTM, achieving nMAE of 4.6%, RE of 27%, and nRMSE of 6.2%. It also excels in the Nash-Sutcliffe metric analysis and Granger causality test, with scores of 0.98, and 0.0992, respectively. Experimental outcomes demonstrate the model's effectiveness in providing accurate wind power predictions, aiding in the efficient management of renewable energy systems and contributing to the advancement of clean energy technology. Authors in [36] propose a Hybrid Deep Learning Model (DLM) for enhancing PV power output forecasting under dynamic environmental conditions. This model combines CNN, LSTM, and Bi-LSTM to capture spatial and temporal dependencies in weather data. Using the Kepler Optimization Algorithm (KOA) for hyperparameter tuning and Transductive Transfer Learning (TTL) for resource efficiency, the model is trained on diverse PV site datasets. Evaluations show the hybrid DLM outperforms individual models in short-term PV power forecasting, demonstrating superior accuracy and resilience, making it effective for PV power plant management. However, there is room for improving the prediction accuracy of solar PV power while ensuring the stability of micro grid operation by investigating more robust deep learning models. Moreover, these studies highlight the potential of integrating the LSTM with different AI architectures to enhance solar power forecasting. However, further research is needed to explore the explanation aspect of the deep learning models as well as optimize the intricate hyper-parameters of the model to improve prediction performance. Table 1 summarizes some of previous works for solar prediction systems based on AI tools.

Responding to the issues raised in the studies in order to boost solar power prediction accuracy and guarantee micro grid operation reliability. This paper proposes a solar power prediction model based on LSTM architecture and EO algorithm, called X-LSTM-EO. The proposed X-LSTM-EO model operates in two stages. The first one employs the LSTM to learn power generation trends based on the environmental conditions and then predict the generating energy, while the second stage which is using the EO algorithm that aims to optimize hyperparameters for the deep learning model, including the number of LSTM cells, the choice of activation function (such as sigmoid, SoftMax, tanh, etc.), and the type of optimizer function (such as Adam, RMSprop, etc.), are all important components used in training a neural network. The proposed X-LSTM-EO scheme is trained and tested with the help of the power plant's PV power output. Because the solar palettes could have lot of issues, Local Interpretable and Model-agnostic Explanation (LIME), an approach for explainable artificial intelligence (XAI), is used to identify the critical conditions for predicting power generation in a smart solar system. The accuracy of the proposed deep learning model was compared and verified with other models in terms of several metrics, including $R^2$, RMSE, COV, MAE, and EC. Results indicate that this approach enhances forecasting accuracy and outperforms the compared models in forecasting efficacy.

The main contribution and the novelty of this paper is summarized as follows:

- Deep learning models might not be as accurate because they use traditional optimization methods to find the best internal parameters. These techniques can get stuck in local minima, which leads to finding parameters that aren't as good as they could be. So, this paper solves these problems by adding the LSTM and the EO algorithm to the proposed model.

**Table 1. Related works for solar power prediction based on AI tools.**

| Ref. | Topic | Technique | Performance measure |
|------|-------|-----------|---------------------|
| [21] | Prediction of hourly day-ahead solar irradiance | LSTM | ▪ Cape Verde dataset: RMSE: 122.7174<br>▪ MIDC dataset: RMSE: 76.245 |
| [22] | Forecasting the output power of PV systems | LSTM-RNN | ▪ Dataset1: RMSE: 82.15<br>▪ Dataset2: RMSE: 136.87 |
| [23] | Estimating the short-term energy output of a PV system | CNN-BiLSTM | ▪ RMSE: 0.056 |
| [24] | Estimating a day ahead solar irradiance | FWPD-BiLSTM | ▪ Monthly forecasting: RMSE: 31.427<br>▪ Seasonal forecasting: RMSE: 13.920 (Winter) RMSE: 45.91 (Monsoon), RMSE: 20.64 (Summer), and RMSE: 15.48 (Autumn) |
| [25] | Estimation of solar irradiance | BiLSTM | ▪ RMSE: 11.35 |
| [26] | Forecasting of solar global horizontal irradiance | RLMD-BiLSTM | RMSE: range from 16.34 to 35.07 |
| [27] | Forecasting PV power generation | GRU | ▪ nRMSE: 0.0292 |
| [28] | Prediction of PV power | LSTM | ▪ RMSE:7.26 |
| [29] | Prediction of solar irradiance | LSTM-BP | ▪ RMSE: 82.716 |
| [30] | Forecasting PV power generation | CNN-LSTM | ▪ RMSE: range from 0.068 to 0.091 |
| [31] | Prediction of solar irradiance | RLSTM | ▪ RMSE: range from 0.05 to 0.14 |
| [32] | Forecasting 24-hour ahead solar GHI | ConvLSTM | ▪ RMSE: range from 0.126 to 0.136 (Spring)<br>▪ RMSE: range from 0.103 to 0.112 (Winter)<br>▪ RMSE: range from 0.129 to 0.150 (Summer)<br>▪ RMSE: range from 0.119 to 0.134 (Fall) |
| [33] | Forecasting PV power generation | CNN-BiLSTM-Attention | ▪ RMSE: 16.1936 (Spring)<br>▪ RMSE: 19.7285 (Summer)<br>▪ RMSE: 25.5638 (Autumn)<br>▪ RMSE: 33.4984 (Winter) |
| [34] | Forecasting PV power generation | FL-TL-Conv-SGRU | ▪ Group1 PV datasets: RMSE: 0.0301 (Summer)<br>▪ Group1 PV datasets: RMSE: 0.0304 (Winter)<br>▪ Group 2 PV datasets: RMSE: 0.0303 (Summer)<br>▪ Group2 PV datasets: RMSE: 0.0305 (Winter) |
| [35] | Prediction of the short-term wind and PV power | COA-CNN-LSTM | vnRMSE:0.062 |
| [36] | Enhancing PV power output forecasting | KOA-CNN-Bi-LSTM | ▪ RMSE:0.0027 |

This model is then used to show correctly how solar output power is related to external factors.

- Applying the EO algorithm for tuning the hyper-parameters of the LSTM to enhance the performance of the forecasting.

- Applying PSO optimizer for comparing its results with EO optimizer.

- The utilization of LSTM for effective exploration of the search space without being trapped in local optima areas.

- To understand the forecasting results. XAI's approach called LIME has been applied to explain the obtained results and performance of the proposed deep learning model.

- The XAI explained the most important environmental condition that affects the model's forecasting results.

- The propped X-LSTM-EO model proposes a common, accurate model that predicts well under many environmental scenarios. It mitigates PV power generation unpredictability and safely integrates large-scale PV power generation into micro grids, lowering operational costs and boosting efficiency and safety.

The following sections outline the rest of the paper. Section 2 provides an overview of the materials and methods used. The dataset description and analysis are illustrated in Section 3. The proposed X-LSTM-EO model is presented in Section 4. Section 5 presents and analyzes the experimental results. Finally, Section 6 summarizes the conclusions and presents the future works.

## 2. Preliminaries

This section provides the basic concepts regarding the LSTM, EO, and Locally Interpretable Model Agnostic Explanations (LIME)

### 2.1 LSTM (Long short-term memory)

The LSTM is categorized as a type of RNN, which is a potent type of artificial neural network that has the capacity to store input data in memory internally. Because of this characteristic, RNNs are particularly effective in addressing problems that involve sequential data, such as time series. However, a major problem that RNNs frequently experience is known as vanishing gradient, which causes the learning process of the model to become extremely slow or even stop altogether [29]. In order to anticipate a time series' future patterns, its previous data is crucial. The time series' historical data is encoded using the LSTM. Long-term memory functionality in the LSTM model may help with long-term sequence modeling's gradient vanishing and exploding issues. We feed the feature vector $x_t$ into the LSTM model at time step $t$. formally; the calculations are carried out by the LSTM model as follows Eqs (1)–(6):

$$f_t = \sigma(W_f[h_{t-1}.x_t] + b_f) \tag{1}$$

$$i_t = \sigma(W_i[h_{t-1}.x_t] + b_i) \tag{2}$$

$$o_t = \sigma(W_o[h_{t-1}.x_t] + b_o) \tag{3}$$

$$\tilde{c}_t = tanh(W_c[h_{t-1}.x_t] + b_c) \tag{4}$$

$$c_t = f_t \odot c_{t-1} + i_t \odot \tilde{c}_t \tag{5}$$

$$h_t = o_t \odot \tanh(c_t) \tag{6}$$

The architecture of the LSTM network includes various components such as input gates, forget gates, output gates, and unit states. A depiction of the network's fundamental structure is presented in Fig 1 [30]. The hidden state in this case, $h_{t-1}$, contains all the data up to the (t -1)[th] time step. Concatenation of $h_{t-1}$ and $x_t$ produces the forget gate $f_t$, input gate $i_t$, and output gate $o_t$, respectively. To create a candidate cell state $\tilde{c}_t$ that symbolizes the newly added information, $h_{t-1}$ and $x_t$ are also employed. Then, $c_t$ is created by combining $c_{t-1}$ and $\tilde{c}_t$, with $f_t$ acting as the procedure's balance factor. To output the current hidden state, $h_t$, $o_t$ is finally multiplied by $c_t$. $W_f$, $W_i$, $W_o$, and $W_c$ are the parameters to be learned, $\odot$ represents the Hadamard product, $\sigma(.)$ and tanh(.) are the sigmoid and tanh activation functions, respectively.

### 2.2 Basics of the EO

EO is a new meta-heuristic method that was suggested by Faramarzi [37] based on physics concept for handling engineering optimization problems. EO simulates the dynamic and equilibrium states that achieve the control volume mass balance models. EO is mathematically

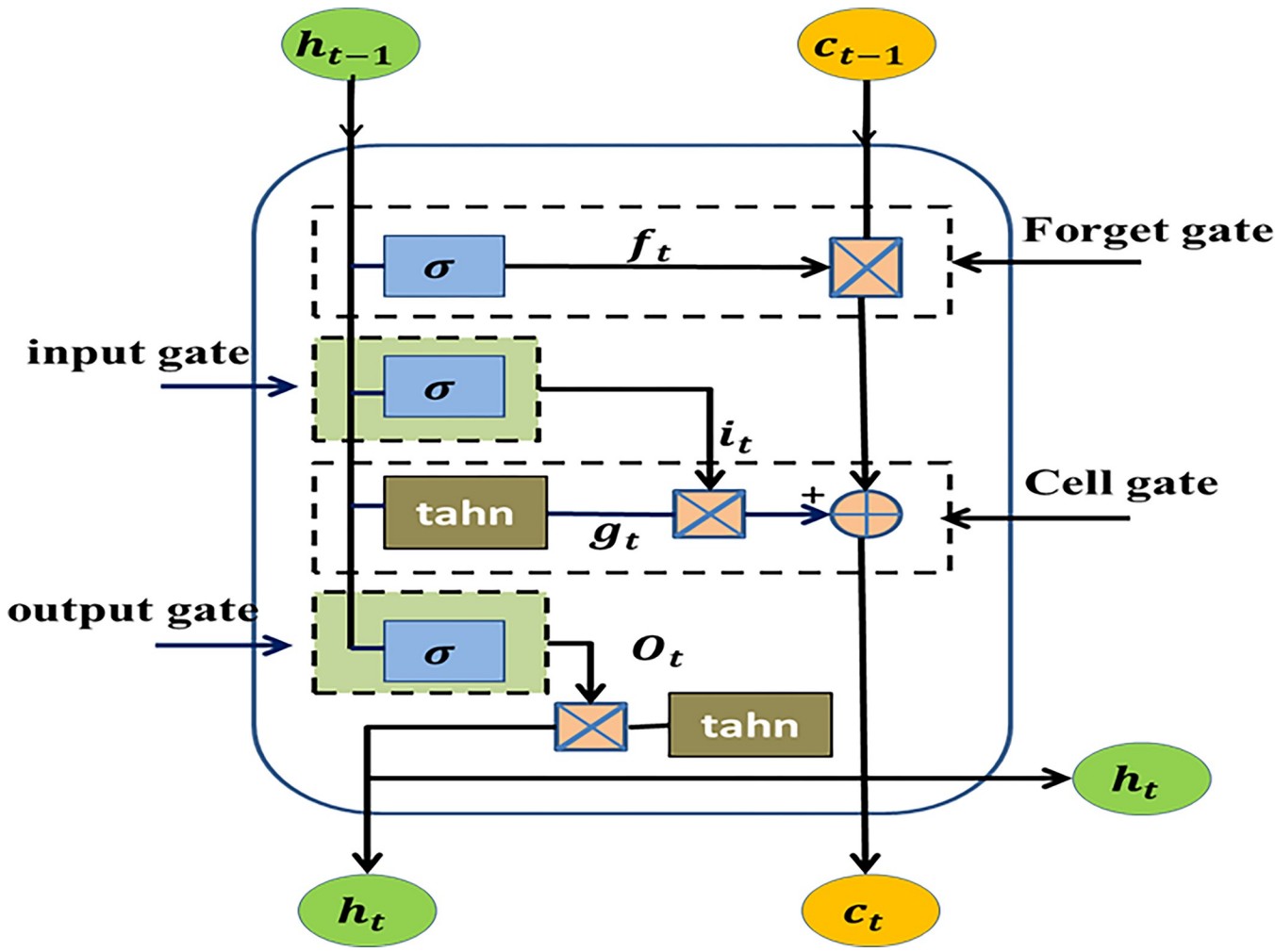

**Fig 1. The LSTM structure.**

composed of search agents to define the particles (solutions) associated with their concentrations (positions). In this regard, the search agent renewed its concentration by choosing one of the best-so-far solutions at random (i.e., equilibrium candidate) to ultimately acquire the optimal outcome (i.e., equilibrium state). Furthermore, EO utilizes the "generation rate" step to boost the search skills in terms of explorative and exploitative trends while avoiding the stuck-in local optima dilemma. The mathematical optimization framework of EO is expressed by the following steps and its algorithm is shown in Algorithm (1).

1. Create an initial population of $N$ particles at random as follows:

$$\Delta_i^{\text{initial}} = \Delta_{\text{lower}} + r_i.(\Delta_{\text{upper}} - \Delta_{\text{lower}}).i = 1.2\ldots.N \qquad (7)$$

Where $N$ denotes population size, $\Delta_{\text{lower}}$ and $\Delta_{\text{upper}}$ denote respectively the lower and upper bounds of the search region, $r_i$ stands for a uniform random vector generated inside the interval [0, 1]. $\Delta_i^{\text{initial}}$ Defines the initial position of the $i^{th}$ particle.

2. Construct the equilibrium candidates ($\bar{\Delta}_{eq.pool}$) as in Eq (8) by adding the best four particles along with their average to $\bar{C}_{eq.pool}$ at aiming to enhance the exploration and exploitation capabilities of EO.

$$\bar{\Delta}_{eq.pool} = \{\bar{\Delta}_{eq(1)}.\bar{\Delta}_{eq(2)}.\bar{\Delta}_{eq(3)}.\bar{\Delta}_{eq(4)}.\bar{\Delta}_{eq(ave)}\} \tag{8}$$

3. Renew the concentration of the particle through following one of the equilibrium candidates chosen at random from $\bar{\Delta}_{eq.pool}$ as follows.

$$\bar{\Delta} = \bar{\Delta}_{eq} + \left(\bar{\Delta} - \bar{\Delta}_{eq}\right).\bar{F} + \frac{\bar{G}}{\bar{\lambda}V}(1 - \bar{F}) \tag{9}$$

Where $\bar{F}$ defines an exponential equation that controls the balance among the explorative and exploitative features, and it is formulated as follows.

$$\bar{F} = a_1.\text{sign}(\bar{r} - 0.5)[1 - e^{-\bar{\lambda}t}] \tag{10}$$

Where $\bar{\lambda}$ defines turnover vector which is composed of random numbers inside the interval [0, 1], $t$ denotes the time which decreases gradually with iterations, and it is expressed as follows.

$$t = \left(1 - \frac{Iter}{Max\_iter}\right)^{\left(a_2\frac{Iter}{Max\_iter}\right)} \tag{11}$$

Moreover, the initial start time ($t_0$) is formulated by the following equation.

$$\bar{t}_0 = \frac{1}{\bar{\lambda}}ln(-a_1.\text{sign}(\bar{r} - 0.5)[1 - e^{-\bar{\lambda}t}]) + t \tag{12}$$

where $Max\_iter$ denotes the maximum iterations, $a_1$ stands for parameter that controls the exploration feature while $a_2$ signifies a parameter that control the exploitation skill, $\bar{r}$ is a vector contains random values ranged from 0 to 1. Additionally, $\bar{G}$ denotes the generation rate parameter that aids to further improve the exploitation feature and it is formulated as follows.

$$\bar{G} = \bar{G}_0 e^{-\bar{\lambda}(t-t_0)} = \bar{G}_0\bar{F}v \tag{13}$$

Where

$$\bar{G}_0 = \bar{GCP}(\bar{C}_{eq} - \bar{\lambda}\bar{C}).\bar{GCP} = \begin{cases} 0.5r_1 & r_2 \geq GP \\ 0 & r_2 < GP \end{cases}$$

where $r_1$ and $r_2$ stand for arbitrary numbers generated using the uniform distribution ranged from 0 to 1; $GP$ denotes the generation probability and it is set to 0.5 to acquire a better balance among the explorative and exploitative skills; $\bar{GCP}$ signifies the generation rate; $V$ is set to a unit.

The concentration update formula is expressed as follows:

$$\bar{\Delta} = \bar{\Delta}_{eq} + \left(\bar{\Delta} - \bar{\Delta}_{eq}\right).\bar{F} + \frac{\bar{G}}{\bar{\lambda} V}(1 - \bar{F}) \tag{14}$$

```
Algorithm 1: Pseudo-code of the proposed EO algorithm
1: Define the algorithm' parameters: a₁ = 2; a₂ = 1; Iter = 0
(counter); GP = 0.5
2: Create an initial population at random comosed of N particles,
{Δ̄ᵢ}ᴹᵢ₌₁
3: For minimization problem, set large value for the fitness of the
equilibrium candidates (Δ̄ₑq,ₚₒₒₗ)
4: While Iter<Max_Iter do
5: For i = 1: N
6: Evaluate the fitness of the particle (f(Δᵢ))
7: If f(Δ̄ᵢ) < f(Δ̄ₑq(₁)), then Δ̄ₑq(₁) ← Δᵢ and f(Δ̄ₑq(₁)) ← f(Δ̄ᵢ)
8: Elseif f(Δ̄ᵢ) > f(Δ) and f(Δ̄ᵢ) < f(Δ̄ₑq(₂)), then Δ̄ₑq(₂) ← Δᵢ and f(Δ̄ₑq(₂)) ← f(Δ̄ᵢ)
9: Elseif f(Δ̄ᵢ) > f(Δ̄ₑq(₁)) & f(Δ̄ᵢ) > f(Δ̄ₑq(₂)) and f(Δ̄ᵢ) < f(Δ̄ₑq(₃)), then Δ̄ₑq(₃) ← Δᵢ and
f(Δ̄ₑq(₃)) ← f(Δ̄ᵢ)
10: Elseif f(Δ̄ᵢ) > f(Δ̄ₑq(₁)) & f(Δ̄ᵢ) > f(Δ̄ₑq(₂)) & f(Δ̄ᵢ) > f(Δ̄ₑq(₃)) and f(Δ̄ᵢ) < f(Δ̄ₑq(₄)),
then Δ̄ₑq(₄) ← Δᵢ and f(Δ̄ₑq(₄)) ← f(Δ̄ᵢ)
11: End If.
12: End for.
13: Δ̄ₑq(ₐvₑ) = (Δ̄ₑq(₁)+Δ̄ₑq(₂)+Δ̄ₑq(₃)+Δ̄ₑq(₄))/4.
14: Construct the equilibrium pool, Δ̄ₑq.ₚₒₒₗ = {Δ̄ₑq(₁).Δ̄ₑq(₂).Δ̄ₑq(₃).Δ̄ₑq(₄).Δ̄ₑq(ₐvₑ)}
15: Renew the time t, t = (1 - Iter/Max_iter)^(a₂ Iter/Max_iter)
16: For i = 1: N
17: Select one equilibrium candidate at random from equilibrium pool
18: Generate the random vectors of λ̄.r̄
19: Update F̄ = a₁.sign(r̄ - 0.5)[1 - e^(-λ̄t)]
20: Perfrom GCP̄ = { 0.5r₁    r₂ ≥ GP
                     0        r₂ < GP
21: Constitute Ḡ₀ = GCP̄(C̄ₑq - λ̄C̄)
22: Constitute Ḡ = Ḡ₀F̄
23: Renew the postion of particle as: Δ̄ = Δ̄ₑq + (Δ̄ - Δ̄ₑq).F̄ + Ḡ/(λ̄V)(1 - F̄)
24: End for
25: Iter = Iter+1
26: End While
27: Output: display Δ̄ₑq
```

## 2.3 Explanation AI based Locally Interpretable Model Agnostic Explanations (LIME)

The technique of "Agnostic Explanations" is a post-hoc, model-agnostic explanation approach that aims to provide interpretability to any black box machine learning model by creating a local, interpretable model for each prediction. LIME is independent of the classifier's algorithm; the authors advise using it to explain any classifier [38]. LIME predicts locally and provides explanations for each observation. LIME fits a local model using similar data points to explain the observation. Linear models, decision trees, and others can be used as local models. The LIME explanation $\varphi(x)$ at the point $x$ produced by an interpretable model $g$ can be

expressed as:

$$\varphi(x) = argmin_{g \in G} L(f.g.\pi_x) + \Omega(g) \tag{15}$$

Where G represents the class of interpretable models. The explanatory model *x*, for example, is a model *g* that minimizes losses like the sum of squared errors. This is the loss *L*, which shows how well the forecasts of the original model *f* can be explained.

$\pi_x$: defines the neighborhood's size in terms of, for example, *x*.

$\Omega(g)$: shows how complex this model is and suggests that the feature amount should be reduced.

The objective is to minimize the locality aware loss function L without making any assumptions about the function *f*, as LIME is designed to be model agnostic. The measure of how accurate g is in approximating *f* in the defined locality is captured by $\pi_x$.

## 3. Dataset description and analysis

In this paper, the data was collected at two solar power plants in India over 34 days [39]. It has a pair of files contains a dataset on power generation and a dataset on sensor readings.

The dataset on Sensor readings includes time and date Observations recorded at 15-minute intervals, it has; "Plant ID", "SOURCE KEY", "AMBIENT TEMPERATURE", "MODULE TEMPERATURE", and "IRRADIATION". The samples of data are listed in Table 2.

While Power-generation data includes Date and time for each observation taken at 15-minute intervals, it has; Plant ID (common to the file), SOURCE KEY sort—The source key in this file represents the inverter id, DC_POWER, AC_POWER, DAILY_YIELD, and TOTAL_YIELD, the samples of data are listed in Table 3.

As noted, the database contains two files which are power generation data and sensor readings data. Table 4 illustrates the statistical analysis of the entire dataset. Scatter plots are used to monitor and describe the relationship between data aspects. Scatter plots show dataset trends and individual data values. These data can establish correlations [40]. Hence, a scatterplot diagram displays dataset data. Scatter plots depict each pair of attributes as dispersion plots. Scatter plots reveal the strongest and weakest associations. This lets us explain each feature's relationship.

Fig 2 shows the solar energy dataset scatterplot graphs. Scatter graphs correlated scatter plots differently. With 23 days' worth of data on solar power generation, the data visualization is used to spot faults and abnormalities in solar power plant output. Fig 3 illustrates that the DC POWER generation per day graph shows that the amount of power made by the sun changes from day to day. On some days, there is less change in how much DC POWER is made. On the other days, the amount of DC POWER produced goes up and down a lot.

The daily DC POWER generation statistic indicates the average daily power generation. Fig 4(A) shows 2020-05-25 has the highest average DC POWER generation and 2020-05-18 the

**Table 2. Weather input data.**

| "DATE_TIME" | "PLANT_ID" | "SOURCE_KEY" | "AMBIENT_TEMPERATURE" | "MODULE_TEMPERATURE" | "IRRADIATION" |
|---|---|---|---|---|---|
| 5/15/2020 3:45 | 4135001 | HmiyD2TTLFNqkNe | 24.8790995 | 23.70979413 | 0 |
| 5/15/2020 4:00 | 4135001 | HmiyD2TTLFNqkNe | 24.6789022 | 22.58994153 | 0 |
| 5/15/2020 4:15 | 4135001 | HmiyD2TTLFNqkNe | 24.3519308 | 21.78364253 | 0 |
| 5/15/2020 4:30 | 4135001 | HmiyD2TTLFNqkNe | 24.0626222 | 21.85252493 | 0 |
| 5/15/2020 4:45 | 4135001 | HmiyD2TTLFNqkNe | 24.0132242 | 22.306315 | 0 |
| ———— | ———— | ———— | ———— | ———— | ———— |

**Table 3. Power generation data.**

| "DATE_TIME" | "PLANT_ID" | "SOURCE_KEY" | "DC_POWER" | "AC_POWER" | "DAILY_YIELD" | "TOTAL_YIELD" |
|---|---|---|---|---|---|---|
| 15-05-2020 06:00 | 4135001 | 3PZuoBAID5Wc2HD | 58 | 5.585714286 | 0 | 6987759 |
| 15-05-2020 06:00 | 4135001 | 7JYdWkrLSPkdwr4 | 58.42857143 | 5.628571429 | 0 | 7602960 |
| 15-05-2020 06:00 | 4135001 | McdE0feGgRqW7Ca | 54.375 | 5.25 | 0 | 7158964 |
| 15-05-2020 06:00 | 4135001 | bvBOhCH3iADSZry | 37 | 3.571428571 | 0 | 6316803 |
| 15-05-2020 06:00 | 4135001 | iCRJl6heRkivqQ3 | 41.85714286 | 4.028571429 | 0 | 7177992 |
| —————————— | —————— | ———————————————— | ————— | ———— | ———— | ———— |

lowest. A system fault or changing weather may explain this large DC POWER generation mismatch. DC POWER days are shown here. Irradiation histograms mirror daily DC power generation. Solar power stations' DC power comes from the sun. Radiation impacts generation. Radiation Fig 4(B) displays the average daily irrigation compared to Fig 4(A). 2020-05-25 has the most radiation, 2020-05-18 the least. DC POWER and IRRADIATION graphs are near-perfect. The sky is cloudless because radiation, solar panel temperature, and ambient temperature are similar (Fig 4(C)). Rain, clouds, and bad weather likely caused this decline. It's unlikely. Since the amount of energy generated from the solar panel is affected by environmental factors, our proposed model not only predicts this amount but also explains the reasons for these findings. The Correlation between variables is a well-known measure of similarity between two random variables. Pearson correlation coefficient ($\rho cc$) is a measure of the degree to which two random variables are dependent on one another [41].

The Pearson correlation coefficient is provided for a pair of variables $x$ with values $x_i$ and $y$ with values $y_i$ by the equation:

$$\rho cc = \frac{co(x.y)}{\sqrt{\sigma^2(x)\sigma^2(y)}} \tag{16}$$

Where $co$ denotes covariance and $\sigma$ denotes variance.

The coefficient $\rho cc$ can take on a value between 1 and +1. Strong positive correlation occurs for values close to +1, strong negative correlation occurs for values close to 1, and no association occurs for values close to 0 [42]. Since Pearson's correlation establishes a straight line of dependence between two variables, linear analysis is assumed when comparing them.

The power_ generation dataset file provides the generated power, whereas the weather dataset file provides the independent attributes used in solar energy prediction. Here, the direction, shape, and magnitude of the dispersion of the data points between the two files' characteristics are used to determine the existence of a relationship.

Four types of power-generated values can be predicted: DC_POWER, AC_POWER, DAILY_YIELD, and TOTAL_YIELD. The $\rho cc$ analysis was executed to identify the outputs that

**Table 4. Statistical data for the dataset.**

| Statistical measure | DC_POWER | AC_POWER | DAILY_YIELD | TOTAL_YIELD | AMBIENT_TEMP | MODULE_TEMP. | IRRAD_IATION |
|---|---|---|---|---|---|---|---|
| count | 45680 | 45680 | 45680 | 45680 | 45680 | 45680 | 45680 |
| mean | 3197.175971 | 312.652679 | 3313.146538 | 6957007.021 | 25.917168 | 31.877975 | 0.236834 |
| std | 4080.448523 | 398.668968 | 3156.100252 | 417238.6436 | 3.55655 | 12.638448 | 0.306316 |
| min | 0 | 0 | 0 | 6183645 | 20.398505 | 18.140415 | 0 |
| max | 14471.125 | 1410.95 | 9163 | 7846821 | 35.252486 | 65.545714 | 1.221652 |

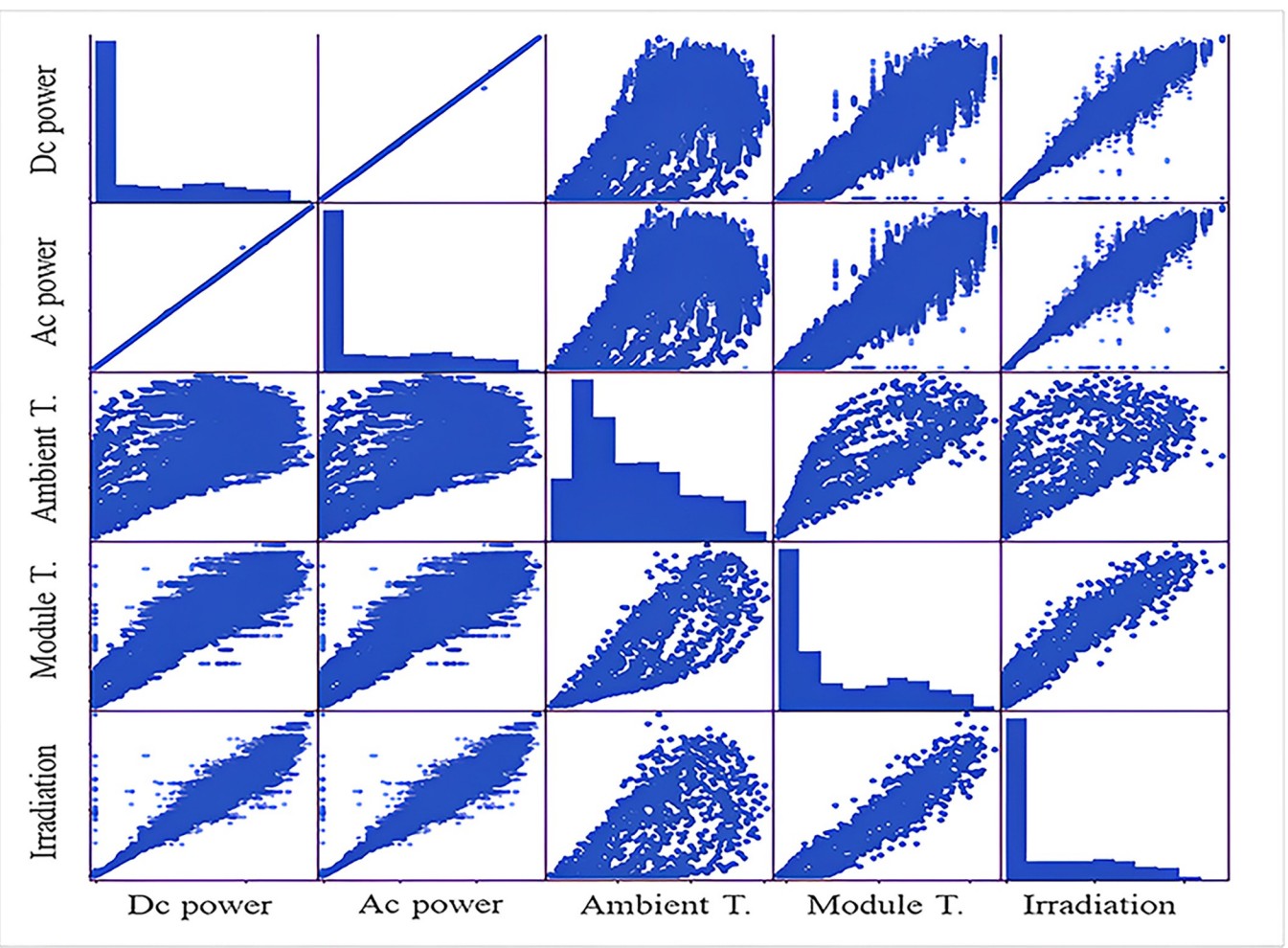

**Fig 2. Data set visualization.**

depended on the input data the most. Table 5 shows the results of the $\rho cc$ analysis; it demonstrates that there is a strong correlation between AC_POWER and DC_POWER; consequently, DC POWER is chosen as the predictability parameter for our work. Since solar panels produce DC power [43].

## 4. The proposed explainable solar power generation forecasting model

The proposed model has four main phases as illustrated in Fig 5 which are data preparation, hyper-parameter optimization, model evaluation, and model explanation. The detailed with each phase will be described in this section.

### 4.1 Data preparation phase

Data preparation involves cleaning and processing raw data for accurate ML predictions. Data preparation, the hardest part of ML, simplifies real-time initiatives. So, in this phase, the data will be aggregated from the two tables: data analysis and visualization and splitting the dataset.

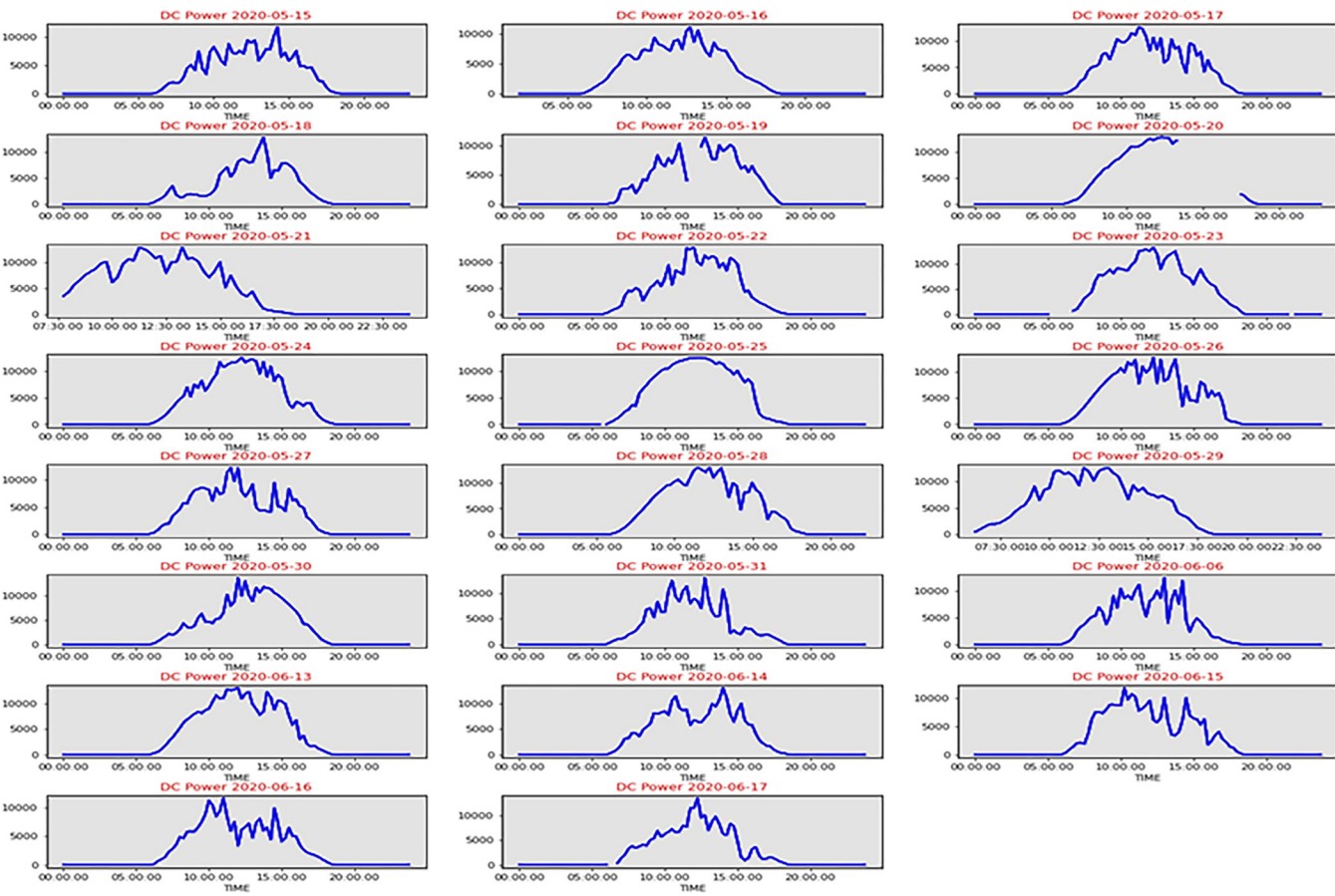

**Fig 3. The variation of DC POWER generation.**

Considering Plant ID, time, and SOURCE KEY; the power generation files; and the sensor reading files combined in a single file, this is done to make the data more conducive to the prediction model, as seen in Table 6.

Splitting a dataset into training and testing sets is a widely used method for model validation. It involves fitting and validating statistics and machine learning models on both the training and testing sets. By allocating a distinct validation dataset, it becomes possible to evaluate and compare the predictive efficacy of different models, mitigating the potential issue of overfitting on the training set. This approach helps to ensure that the model's performance is not biased towards the training data and can generalize well to new data [44].

To effectively analyze a dataset, it is necessary to divide it into a training set comprising n rows and a testing set comprising m rows, where the total number of rows is denoted by Rs = n + m. The splitting ratio, represented by m/Rs, is denoted by the symbol ?. In datasets that contain predictor variables x and u, both of which are less than Rd, the training set is defined as $DS_{train} = \{(xi, yi)\}$, where I = 1,. . ., n, and the testing set is defined as $DS_{test} = \{(ui, vi)\}$, where I = 1,. . ., m. If a predictor variable is categorical, it is presumed to be coded to a numerical variable.

The objective is to develop a model g(x; β) that approximates E(y |x), where β represents a collection of unknown parameters in the model. The training set is utilized to estimate the value of β, while the testing set is used to evaluate the approximation error. To determine the

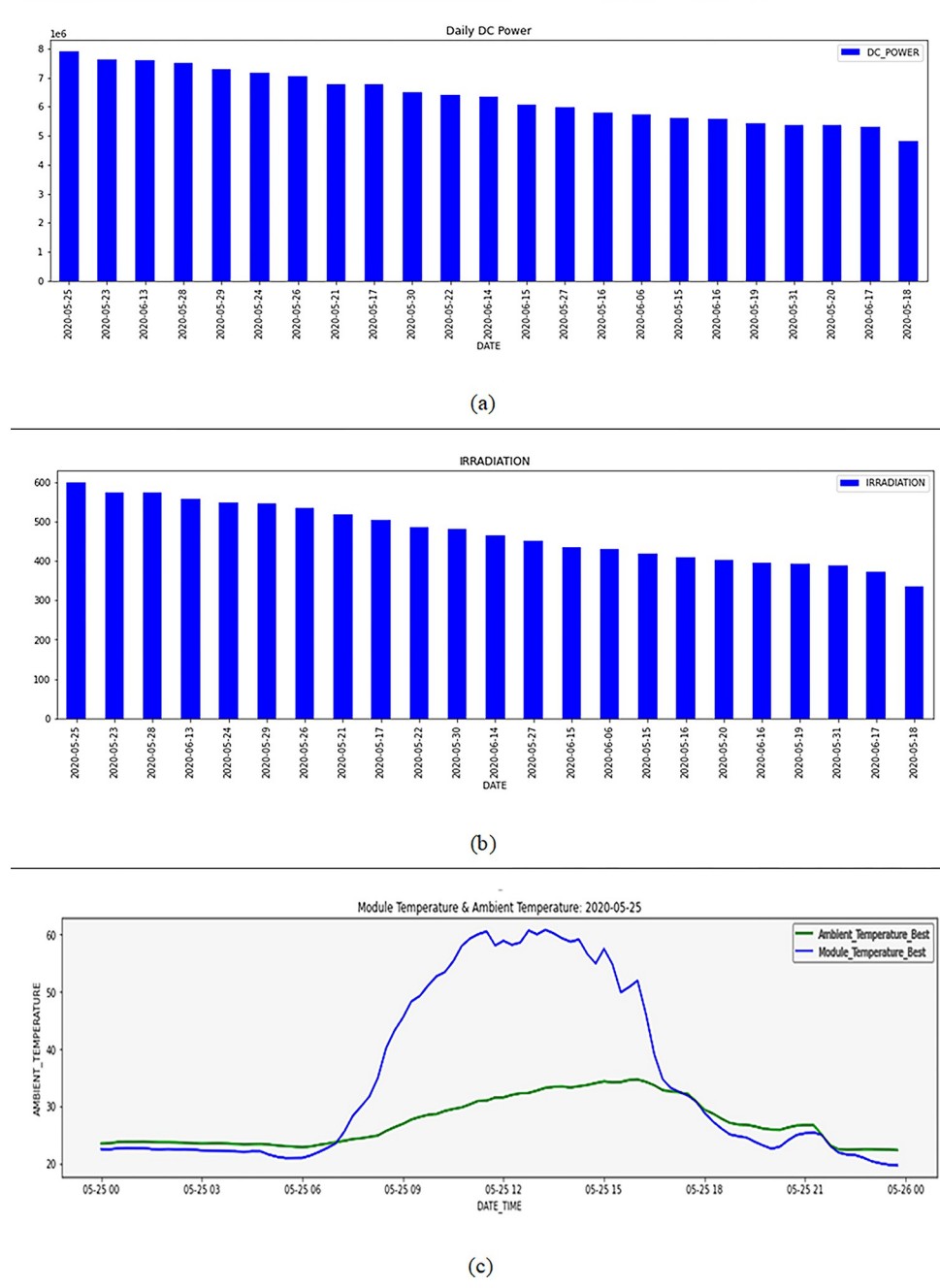

**Fig 4.** daily power tracing, (a) daily DC power, (b) daily irradiation, (c) daily module temperature and ambient temperature.

**Table 5. Pearson correlation coefficient ($\rho cc$) between the input features ad different outputs.**

|  | "DC_POWER" | "AC_POWER" | "DAILY_YIELD" | "TOTAL_YIELD" |
|---|---|---|---|---|
| **AMBIENT_TEMPERATURE** | 0.703795653 | 0.704034933 | 0.489708994 | -0.036531571 |
| **MODULE_TEMPERATURE** | 0.954691732 | 0.95480979 | 0.203702205 | -0.00498142 |
| **IRRADIATION** | 0.991304905 | 0.991259647 | 0.071937367 | -0.00498142 |

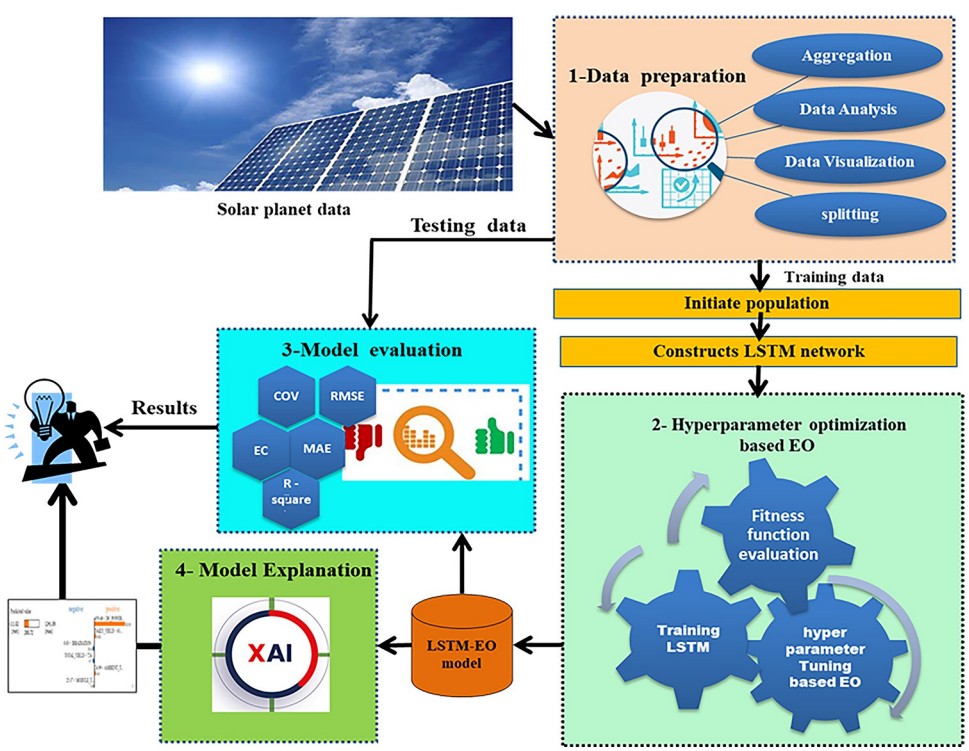

**Fig 5. The proposed X-LSTM-EO model.**

prediction error of the estimated model g(x; β) derived from the training set, a loss function L denoted by L(y, g(x; β)) is used [44].

## 4.2 LSTM hyper-parameter optimization based on EO

Like other neural network models, LSTM applications require manual setting of model parameters such as the number of cells, model optimization function, and training epochs. The network model's performance depends on these factors. This research proposes using the EO method to optimize the internal parameters of LSTM and then using the optimized neural network to predict power generation.

In order to expedite the construction of LSTM models, it is essential to investigate hyperparameter tuning and to generate recommendations that can serve as a solid starting point for the type of network being developed. To get the ideal values for the Hyper-parameters (the numbers of LSTM cells $P1_c$, the numbers of epoch's $P2_e$, and optimization function 's type $P3_o$), the interval values for each parameter are set as shown in Table 7.

**Table 6. Dataset after aggregation.**

| SOURCE_ KEY | DC_ POWER | AC_ POWER | DAILY_ YIELD | TOTAL_ YIELD | DATE_ TIME | AMBIENT_ TEMP | MODULE_ TEMP | IRRAD_ IATION |
|---|---|---|---|---|---|---|---|---|
| VHMLBKoKgIrUVDU | 0 | 0 | 8172 | 7321059 | 5/29/2020 22:15 | 24.67018 | 23.41753 | 0 |
| 3PZuoBAID5Wc2HD | 2011.714286 | 197.185714 | 141.857143 | 7094120.86 | 5/29/2020 7:15 | 21.8082 | 26.07005 | 0.1339 |
| 1BY6WEcLGh8j5v7 | 0 | 0 | 0 | 6455679 | 6/13/2020 2:30 | 21.82477 | 19.7227 | 0 |
| uHbuxQJl8lW7ozc | 253.625 | 24.4625 | 7301.5 | 7267832.5 | 6/14/2020 18:15 | 25.1361 | 24.72327 | 0.01903 |
| 1IF53ai7Xc0U56Y | 9079.75 | 888.175 | 1527.375 | 6242061.38 | 5/23/2020 9:30 | 27.64304 | 47.78662 | 0.61746 |

**Table 7. Hyper-parameters setting interval.**

| Hyper-parameter | Interval |
| --- | --- |
| $P1_c$ | [10,25] |
| $P2_e$ | [25,100] |
| $P3_o$ | [0,1] |

The flow of the proposed model phases for power forecasting is depicted in Fig 6, and the subsequent sections will provide a description of each stage.

**4.2.1 Initialization.**   Particles are used in this stage of EO, with each particle standing in for the concentration vector that holds the optimal solution. In order to generate a random vector of initial concentrations within the search space, Eq 7 is employed. Our particles are denoted by the symbols $P1_c$, $P2_e$, and $P3_o$; their values are denoted by the ranges [10 – 25], [25 – 100], and [0 – 1] (where 0 indicates to Adam optimizer while 1 indicates to SGD optimizer) respectively. Moreover, their initial value is determined by Eq (7).

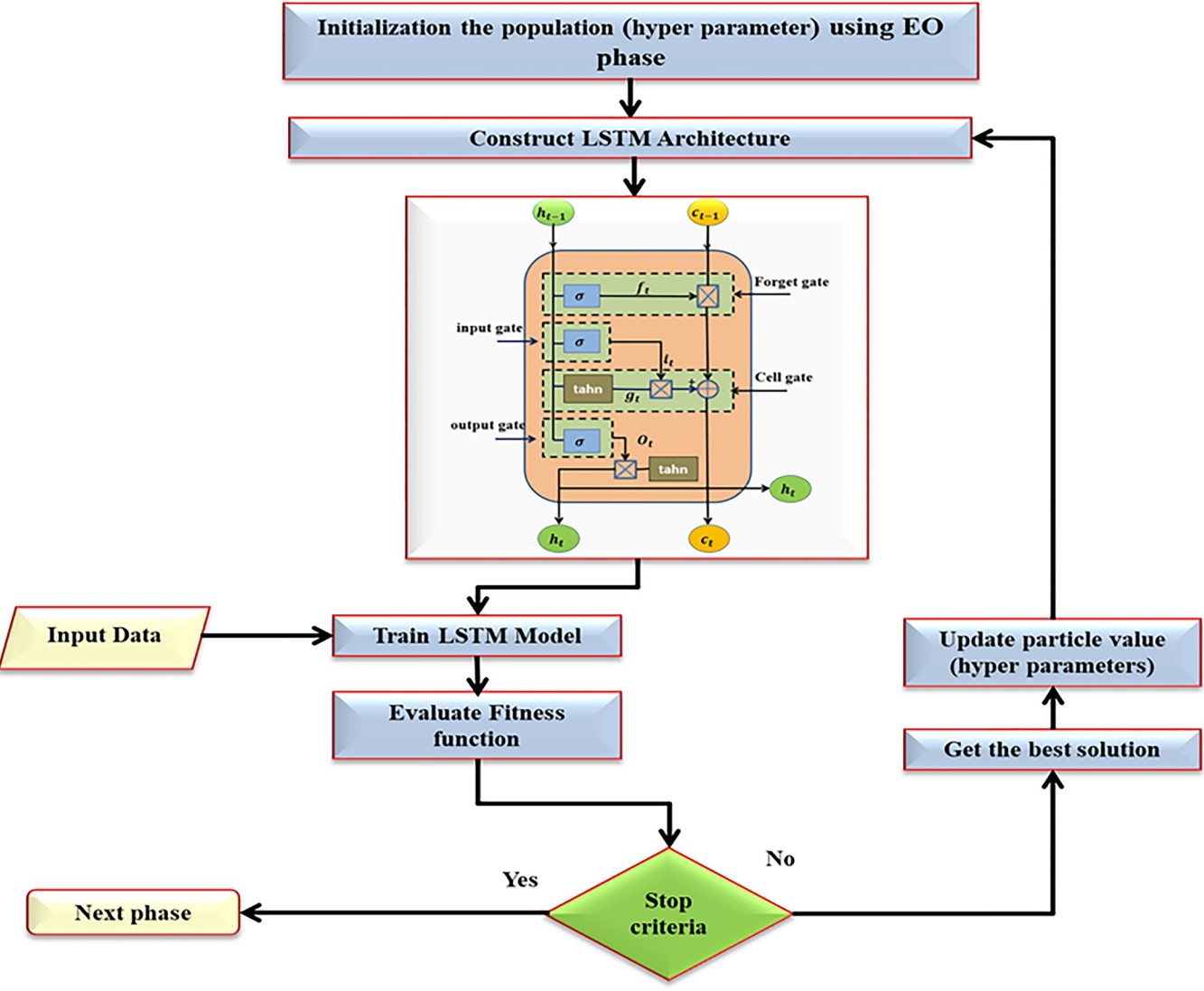

**Fig 6. The flowchart of the LSTM based EO for power forecasting.**

Every meta-heuristic algorithm has a specific objective that aligns with its inherent characteristics. The objective of EO is to attain a state of equilibrium, and by achieving this state, EO can potentially solve the optimization problem with near-optimal results. However, during the optimization phase, the concentration levels required to reach the equilibrium state remain unknown to EO. Once the population reaches equilibrium, the top four particles are analyzed and chosen as potential candidates, thus being added to a candidate list. Additionally, another candidate is chosen based on the average of the top four particles. This list of five equilibrium candidates serves as a valuable tool for EO, allowing it to explore and exploit opportunities effectively. The first four candidates help EO enhance its ability to diversify and exploit on average, while all five candidates are stored in an equilibrium pool vector, as shown in Eq (8).

**4.2.2 Constructing and training LSTM.** Training a model is the process of feeding artificial data to a parametrized machine learning algorithm in order to generate a model with optimal learned trainable parameters that minimize an objective function. At this phase, Data is supplied in batches to LSTM. In the previous phase, the parameters of the LSTM are defined; in the subsequent phase, the performance is evaluated. Fig 7 shows the LSTM Architecture.

**4.2.3 Evaluating fitness function.** The fitness function is a crucial component of all meta-heuristic methods. For solving hyper-parameter tuning problems using a meta-heuristic, EO must be provided till it permits the search core of the optimization process to be identified.

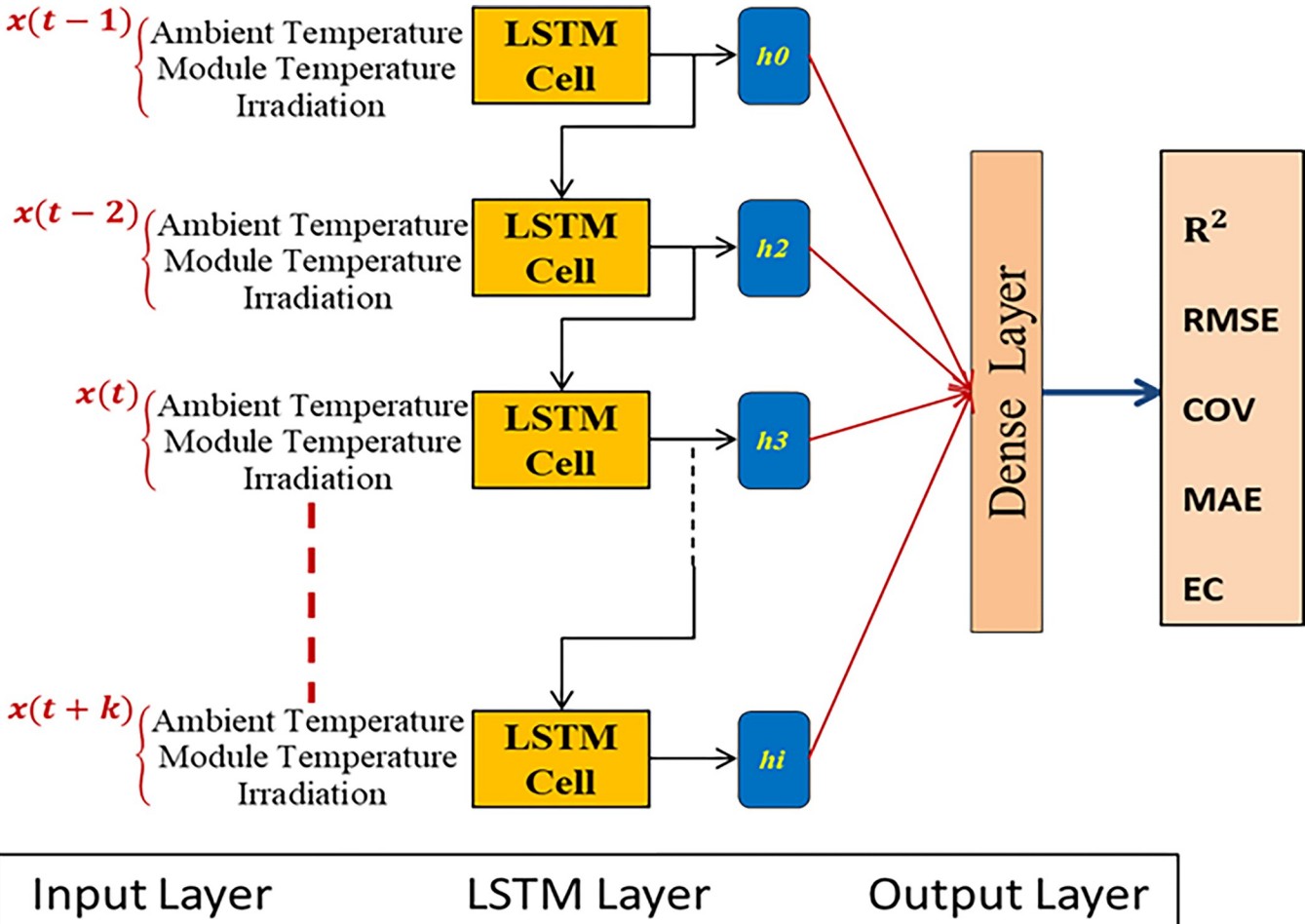

**Fig 7. LSTM architecture.**

The fitness function utilized by EO to solve the hyper-parameter tuning problem is computed by minimizing the MSE of the training process. $\hat{y}_i$ is the output of training process of LSTM model. The LSTM model has a number of LSTM cells $P1_c$ and the optimization algorithm $P3_o$.

$$\hat{y}_i = \text{LSTM}\left(P1_c, P3_o, DS_{\text{train}}(x_i, y_i)\right) \tag{17}$$

$$MSE_{lstm} = \left(\frac{1}{P2_e}\right) \sum_{i=1}^{P2_e} (\hat{y}_i - y_i)^2 \tag{18}$$

$$\text{Fitness function} = argMin\left(MSE_{lstm}\right) \tag{19}$$

**4.2.4 Updating concentration.** The concentration is modified according to Eq 9 based on Eq (18). Where Eq (19) calculates the fitness value based on the newly obtained concentration and then updates the individual ideal concentration and the global optimal concentration of the particles. Fig 8 shows the updated values of the three parameters across the EO optimizer progress and its effects on RMSE values.

### 4.3 Model evaluation

Understanding the strengths and shortcomings of a machine learning model requires evaluating the model using a variety of evaluation measures. In addition to its role in model

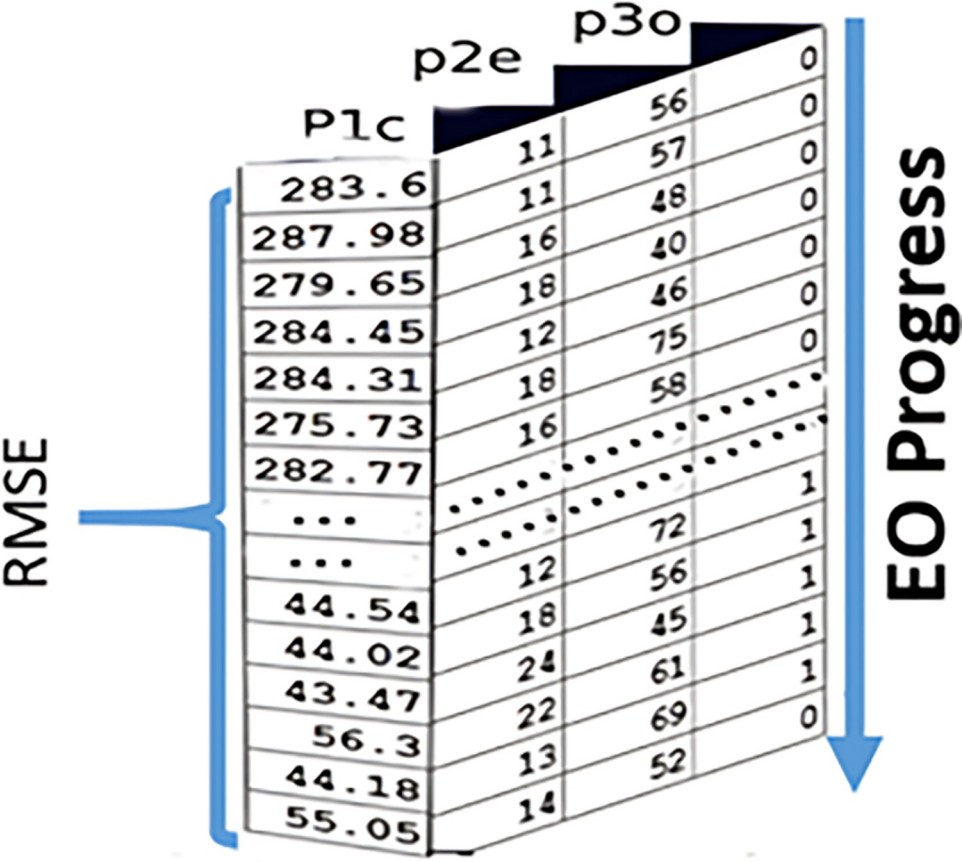

**Fig 8. Updating concentration.**

monitoring, model evaluation is crucial for determining a model's efficacy at the early stages of research.

Numerous statistical metrics are used to evaluate the precision of models created to forecast the performance of solar energy generation. These criteria largely concentrate on quantifying the disparities between the projected values and the real measurements. The statistical methods used for this purpose include the coefficient of determination $R^2$, RMSE, MAE, COV, and EC. The Eqs (20)–(24) mathematically establish the link between these statistical factors, as demonstrated in the reference [45].

**$R^2$**: the R-square's primary goal is to calculate the correlation between predicted and actual data. A dataset contains n values designated as $y_1$, $y_2$,. . .,n (also known as $y_i$ or as a vector y = $[y_1, y_2,. . .,n]^T$), each of which is associated to a forecasted value $f_1$,. . ., $f_n$. It is changed between its lowest value (zero) and its maximum value (one). The better the correlation and the better the artificial model, the closer "$R^2$" is to one.

$$R^2 = \frac{\left(\sum_{i=1}^{n}(d_i - \bar{d})(y_i - \hat{y}_i)\right)^2}{\sum_{i=1}^{n}(d_i - \bar{d})^2 \times \sum_{i=1}^{n}(y_i - \bar{y})^2} \qquad (20)$$

**MAE:** the mean absolute error is determined as the average absolute error of datasets, and it is calculated using Eq 20:

$$MAE = \frac{1}{N}\sum_{i=1}^{N}|d_i - y_i| \qquad (21)$$

**RMSE**: the root mean square error metric is commonly utilized to measure the error between the predicted and measured values, especially for assessing the difference between the predicted and target datasets. As the RMSE value becomes smaller, the accuracy of the model improves.

$$RMSE = \sqrt{\left(\frac{1}{N}\right)\sum_{i=1}^{N}(d_i - y_i)^2} \qquad (22)$$

**COV:** the coefficient of variation is a statistical measure that indicates how far individual datasets in a set of data deviate from the average value. For the model to be more accurate, it needs to have a low coefficient of variation (COV).

$$COV = \left(\sqrt{\left(\frac{RMSE^2}{y^2}\right)} \times 100\right) \qquad (23)$$

**EC:** the efficiency coefficient value is a statistical indicator that determines the accuracy of the model. To ensure that the model is fitted correctly, it is necessary that the EC be equal to 1.

$$EC = 1 - \frac{\sum_{i=1}^{N}(d_i - y_i)^2}{\sum_{i=1}^{N}(d_i - \bar{d})^2} \qquad (24)$$

Where d, and y stand for the measured and predicted values, respectively, and $N$ stands for the number of iterations. $y_i$ is the predicted datasets, and $\bar{d}$ is the mean.

## 4.4 Model explanation based on LIME

The goal is to get a better understanding of how to apply XAI techniques to solar power generation forecasts and how to interpret "black box" machine learning models for usage in solar power station applications. In this paper, the Long-Short Memory (LSTM) is assumed to be

**Table 8. Experiment parameters for solar power generation forecasting using LSTM.**

| Parameter | Value |
|---|---|
| No. of LSTM cell | 10 |
| Optimizer | Adam |
| Epochs | 25 |
| Learning rate | 0.1 |
| batch size | 10 |

the primary black-box model. The preceding section outlined the process of training the LSTM model. To achieve optimal performance, the LSTM model's hyper-parameters are optimized with the EO algorithm. The LIME tool provides aid in determining an interpretable model as part of an interpretable representation at the level of the neighborhood.

## 5. Experiments, results and discussion

This section demonstrates the results of all the experiments carried out to evaluate the efficiency of the proposed model. TensorFlow and Keras, along with GPU processing from Google Collab, were used to analyze the results of all of the experiments.

The LSTM network receives data in batches, and the batch size, which determines the number of rows processed by the model before updating its weights, is determined by the user (in our case, it was set to 10). While processing a batch, LSTM retains its state, and between batches, the state can either be maintained or cleared. By default, the state is cleared. The experiment was conducted using the parameters listed in Table 8.

The experiment involved setting the number of LSTM cells to 10 and using "mean_squared_error" as the loss function for LSTM, along with ADAM as the optimization algorithm and a learning rate of 0.1. The training was conducted with a batch size of 1, and testing was conducted with a batch size of 1 as well. The number of epochs was kept constant at 25.

Table 9 illustrates the experiment test results for solar power generation forecasting using LSTM, where the $R^2$, RMSE, COV, MAE and EC are 0.67, 2.2, 1.31, 1.15 and 0.71 respectively.

The LR, the DT and Gradient Boosting were executed to verify the efficiency of the LSTM; Table 9 shows the experiment results for solar power generation forecasting with different ML algorithms and LSTM.

In Table 9, the LR train results are 1, 1.16, 1.72, 0.00 and 0.92 for $R^2$, RMSE, COV, MAE and EC respectively. While the LR tests results are 1, 0.96, 1.58, 0.18 and 0.86 for $R^2$, COV, MAE and EC respectively. The DT train results are 1, 0.99, 1.8, 0.00 and 1.1 for $R^2$, RMSE, COV, MAE and EC respectively. While the DT tests results are 1, 2.12, 1.1, 1.01 and 0.71 for $R^2$, RMSE, COV, MAE and EC respectively. The Gradient Boosting train results are 1, 0.99, 1.8, 0.00 and 1.1 for $R^2$, RMSE, COV, MAE and EC respectively. While the Gradient

**Table 9. The experiment test results for solar power forecasting using LSTM.**

| Measure | Training results | | | | Testing results | | | |
|---|---|---|---|---|---|---|---|---|
| | LSTM | Decision tree | Linear regression | Gradient Boosting | LSTM | Decision tree | Linear regression | Gradient Boosting |
| $R^2$ | 0.68 | 1 | 1 | 1 | 0.67 | 1 | 1 | 1 |
| RMSE | 2.259 | 0.99 | 1.16 | 2.12 | 2.27 | 0.98 | 0.96 | 2.07 |
| COV | 1.01 | 1.80 | 1.72 | 1.1 | 1.31 | 1.77 | 1.58 | 1.52 |
| MAE | 1.14 | 0.00 | 0.00 | 1.01 | 1.15 | 0.18 | 0.18 | 0.86 |
| EC | 0.801 | 1.1 | 0.92 | 0.71 | 0.71 | 1.01 | 0.86 | 0.64 |

**Table 10. Training time for solar power forecasting.**

| Decision tree training time | Linear regression training time | LSTM training time |
|---|---|---|
| 4.84E-05 min | 1.47E-05 min | 1.41E-01min |

Boosting tests results are 1, 2.07, 1.52, 0.86 and 0.64 for $R^2$, RMSE, COV, MAE and EC respectively.

LR gets better results for $R^2$, RMSE, and MAE measures, but it does not achieve good results for COV and EC; LSTM, on the other hand, does. Therefore, it is proposed that LSTM be used, and attempts are made to improve its efficiency through the use of an optimizer technique.

Table 10 displays the calculation of the training time for the three examined models throughout the training process which indicates the complexity of each one. The training running time for DT, LR, and LSTM are 4.84E-05 min, 1.47E-05 min, and 1.41E-01 min respectively. As demonstrated, LSTM has a longer processing time which means that it is a more complicated one, but it is worthwhile to utilize and strive to enhance its performance due to its superior efficacy.

Fig 9 depicts the actual value and the predicted value of the training model for the LSTM algorithm, as well as the actual value and the predicted value of the testing phase.

The experiment involving the application of the EO algorithm included several important hyper-parameters in the construction of LSTM, such as the training optimizer, training epoch, and number of LSTM cells. The primary training optimizers used were Gradient Descent (SGD) and Adam optimizers. Gradient Descent is an optimization algorithm that follows the negative gradient of an objective function to find its minimum. However, a fixed step size (learning rate) across all input variables is a limitation of Gradient Descent. On the other hand, Adam, short for "Adaptive Moment Estimation," is an extension of Gradient Descent that automatically adjusts the learning rate for each input variable based on the objective function. It further smooths the search process by utilizing an exponentially decreasing moving average of the gradient to update variables [46], the lower and upper order of each parameter is illustrated earlier in Table 7. For the P3$_o$, "0" indicates ADAM optimizer while "1" indicates gradient optimizer.

The EO algorithm was assigned specific parameter values. The maximum number of iterations was set to 50, while a1 was assigned a value of 3 and a2 was assigned a value of 1, which are both recommended values. The values of these parameters are summarized in Table 11.

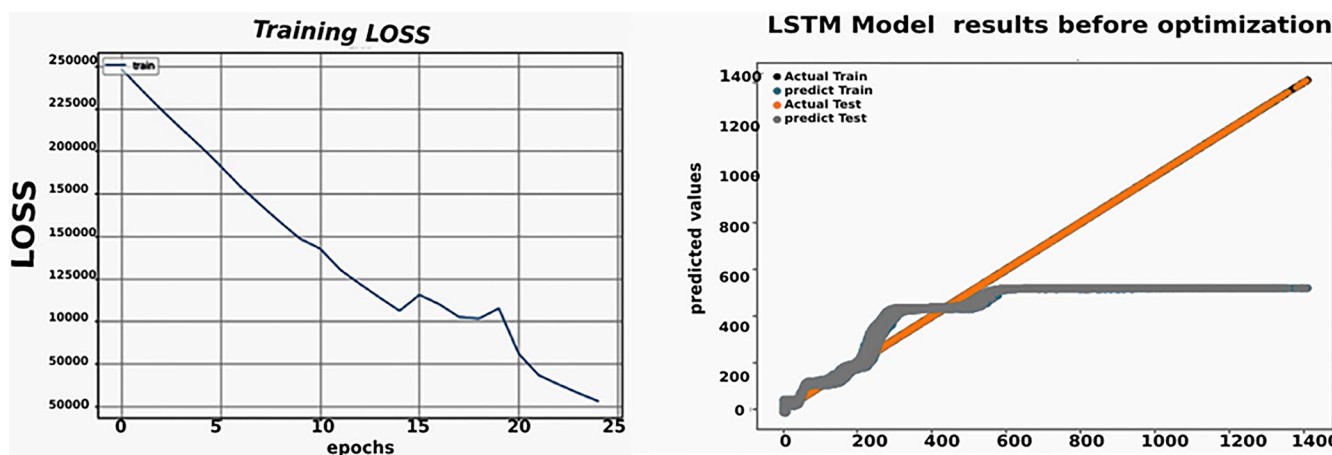

**Fig 9. LSTM performance and results before optimization.**

**Table 11. EO algorithm parameters.**

| Parameter | Value |
|---|---|
| **Maximum iterations** | 50 |
| **Number of particles** | 3 |
| **a1** | 3 |
| **a2** | 1 |

Fig 10 illustrates the EO optimizer execution that occurs while the optimizer is being performed. A sampling of the results of the EO can be shown in Fig 11, which covers a few different epochs.

After applying the EO algorithm, the best solution obtained includes 24 LSTM cells, "MSE" as the LSTM loss function, ADAM as the optimization algorithm with a learning rate of 0.1, a training batch size of 10, and a testing batch size of 10. The number of epochs used for training was fixed at 45.

Table 12 illustrates the Experiment of test results for LSTM based EO, where the $R^2$, RMSE, COV, MAE and EC are 0.99, 0.46, 0.35, 0.229 and 0.95 respectively. Fig 12 depicts the actual value and the predicted value of the training model, as well as the actual value and the predicted value of the testing phase.

The PSO algorithm is utilized to optimize the hyper-parameters of LSTM. The primary objective is to identify the most significant parameters that influence the LSTM's performance and therefore, the PSO algorithm has been implemented to discover the optimal parameters. The PSO is among the most popular metaheuristics. This method was inspired by natural swarm behavior, such as bird flocking and schooling. PSO has been widely used, and it has inspired a new study field known as swarm intelligence [47].

Table 13 illustrates the Experiment of solar power generation forecasting using LSTM based PSO test results, where the $R^2$, RMSE, COV, MAE and EC are 0.9, 0.46, 0.35, 0.229 and 0.95 respectively. The training and testing model's actual values and predicted values are illustrated in Fig 13.

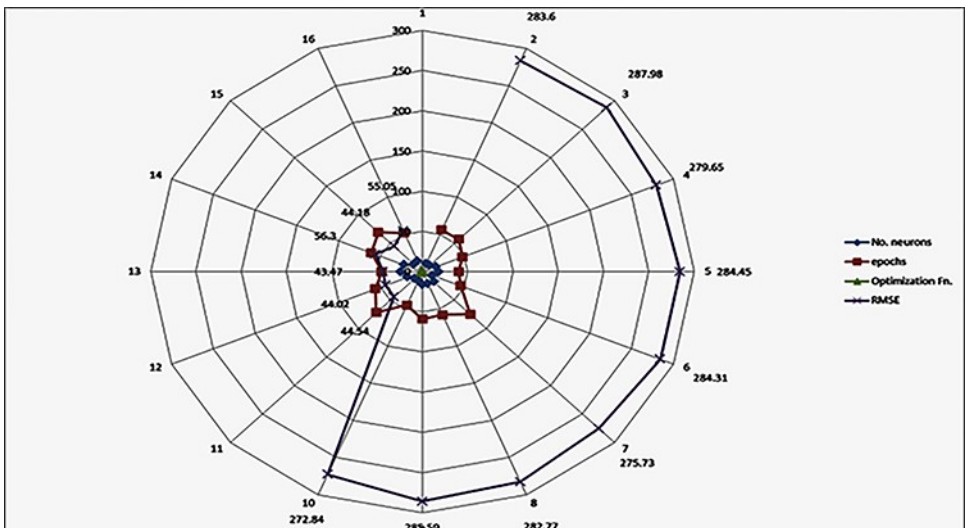

**Fig 10. EO execution performance.**

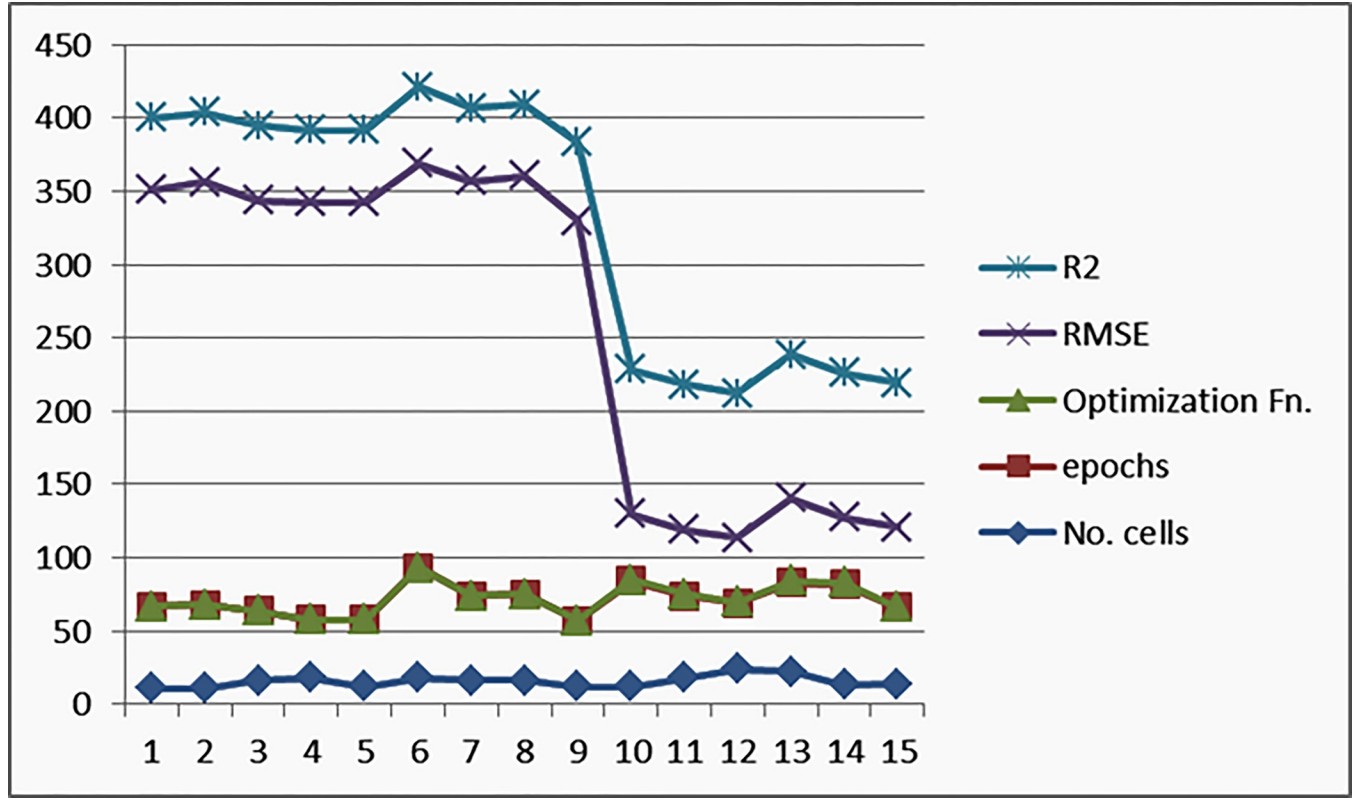

**Fig 11. EO epochs results.**

After implementing the PSO optimization, it was discovered that the loss of the trained model is less when using the EO optimizer than when using the PSO optimization. In Fig 14, the red rectangle above the model's performance covers the 20 to 40 epoch intervals after

**Table 12. The experiment of test results for LSTM based EO.**

| Measure | LSTM Training | LSTM Testing |
|---|---|---|
| $R^2$ | 0.99 | 0.99 |
| RMSE | 0.47 | 0.46 |
| COV | 0.34 | 0.35 |
| MAE | 0.23 | 0.229 |
| EC | 0.96 | 0.95 |

**Table 13. Experiment results of solar power generation forecasting using LSTM based on PSO.**

| Measure | LSTM Training results | LSTM Testing results |
|---|---|---|
| $R^2$ | 0.90 | 0.90 |
| RMSE | 1.25 | 1.27 |
| COV | 0.64 | 0.71 |
| MAE | 0.60 | 0.61 |
| EC | 0.901 | 0.89 |

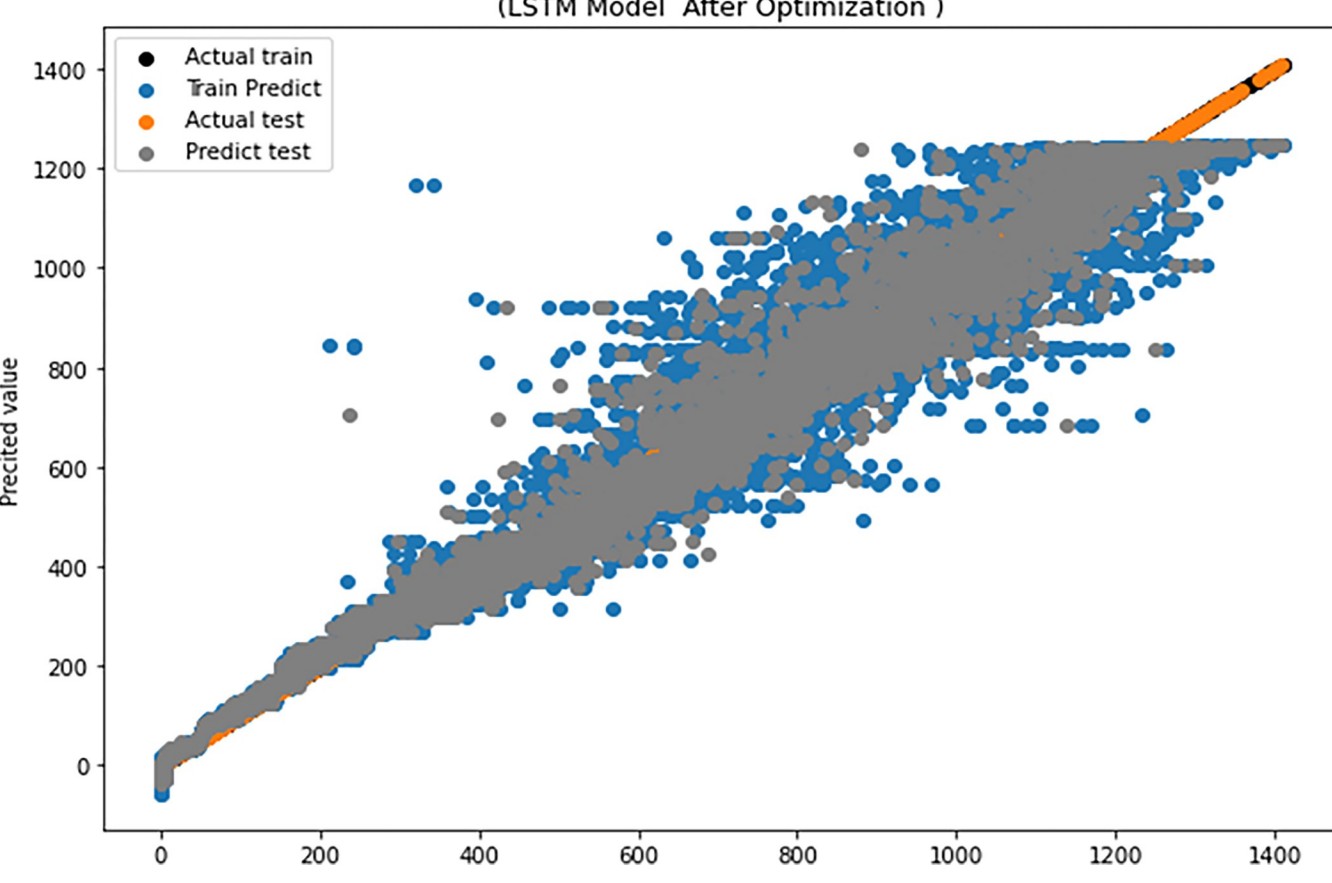

**Fig 12. The LSTM performance after optimization.**

employing PSO, which is offset at the bottom by a second red rectangle covering the model's performance at the same interval after employing EO. This indicates the efficacy of the model based on EO, as the loss is less. It also demonstrates that the performance of the EO-based model is stable and does not fluctuate, as is the case with the PSO-based model.

**Table 14. Comparing with other works.**

| Reference | Method | Model function | Results | Data source |
|---|---|---|---|---|
| [48] | LSTM | prediction of solar power output | MAE: 8.32 RMSE: 19.68 | PV in HOKKAIDO |
| | PSO -LSTM | | MAE: 8.19 RMSE: 19.56 | |
| [49] | LSTM-CNN | Photovoltaic power forecasting | MAE: 0.221 RMSE: 0.621 | 1B DKASC, Alice Springs PV system data |
| [50] | CNN-LSTM | photovoltaic power prediction | MAE: 0.126 RMSE: 0.343 | the 1B DKASC, Alice Springs PV system data |
| [51] | Hybrid KNN-SVM machine learning | solar power forecasting | $R^2$: 98% | Meteonorm provided Jodhpur real-time series dataset from weather station data centers. |

*(Continued)*

**Table 14.** (Continued)

| Reference | Method | Model function | Results | Data source |
|---|---|---|---|---|
| *The proposed hybrid model in this paper* | LSTM | Solar Power Generation Forecasting | $R^2$: 0.67 RMSE: 2.27 COV: 1.31 MAE: 1.15 EC: 0.71 | two solar power plants in India over the course of 34 days |
| | LSTM based PSO | | $R^2$: 0.9 RMSE: 1.27 COV: 0.71 MAE: 0.61 EC: 0.89 | |
| | LSTM based EO | | $R^2$: 0.99 RMSE: 0.46 COV: 0.35 MAE: 0.229 EC: 0.95 | |

Fig 15 depicts the results of the executed experiments, where the best results are the blue-colored LSTM-based EO results; the accuracy presented by EC and $R^2$ is the best, whereas the loss presented by MAE, RME, and COV is the least. While using LSTM gives the worst results (colored with red); the accuracy presented by EC and $R^2$ is the worst, whereas the loss

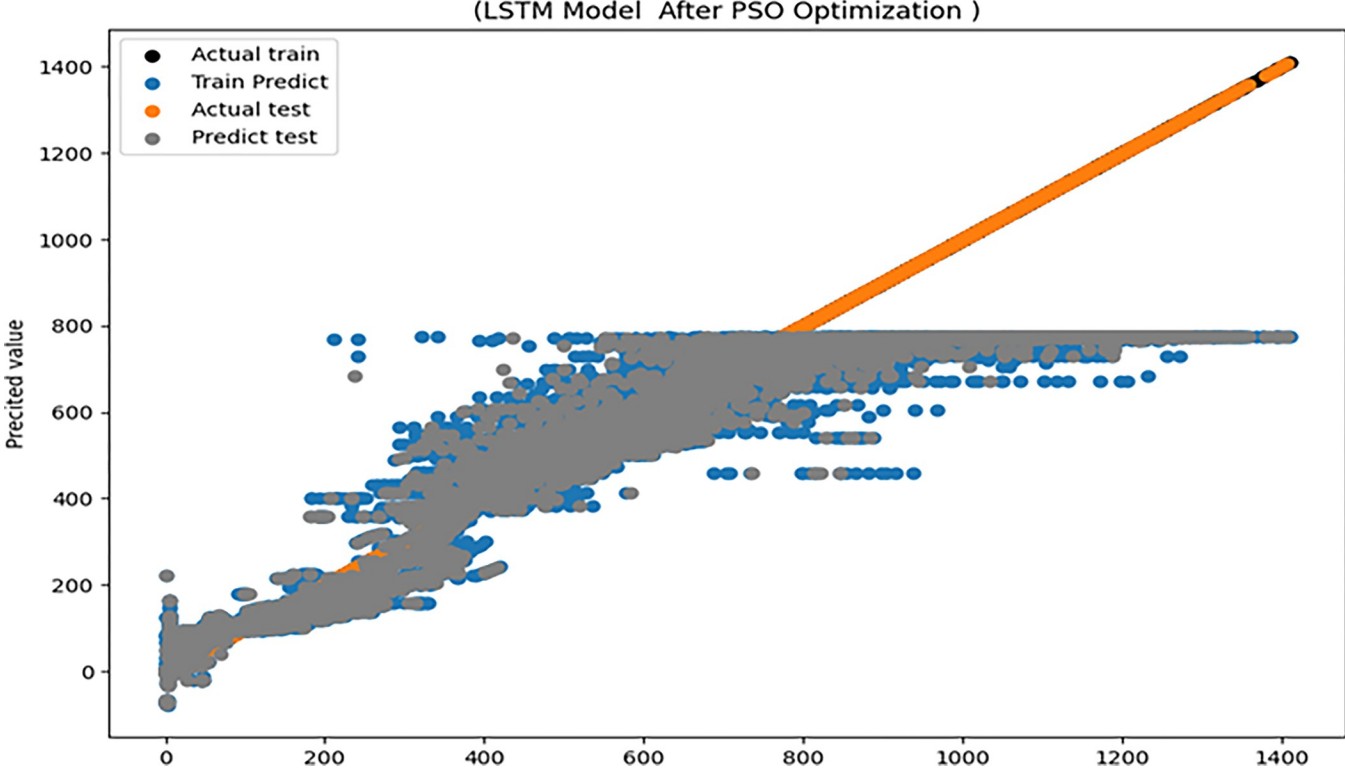

**Fig 13. The LSTM performance based on PSO optimizer.**

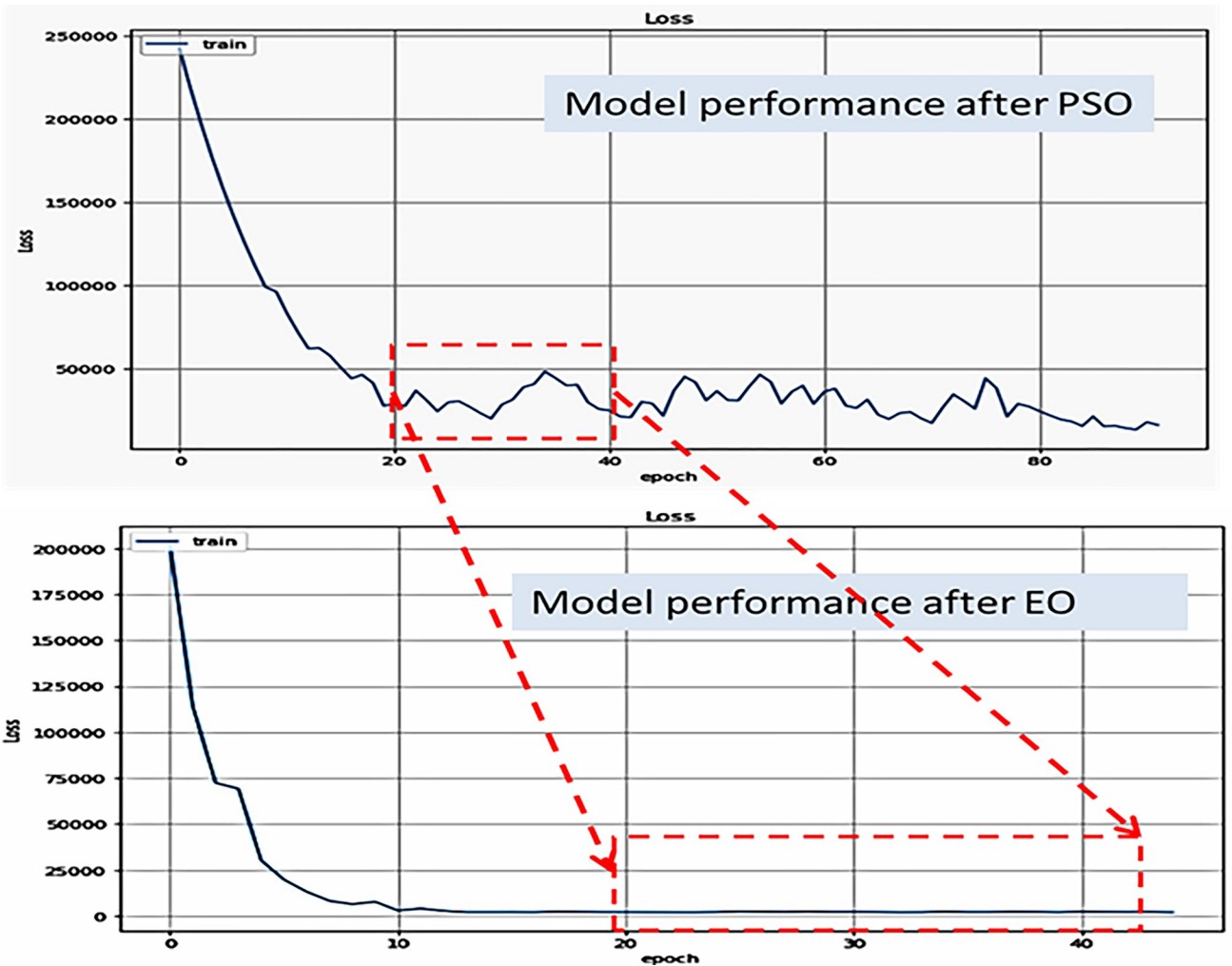

**Fig 14. The comparison of LSTM performance based EO vs. PSO optimizers.**

presented by MAE, RME, and COV is the highest. LSTM-based PSO results (colored with green) give moderated results.

## 5.1 Comparative analysis with other studies

To test and confirm the correctness of the proposed model, it was compared with the work of others who have employed LSTM. Table 14 demonstrates that the proposed model achieves superior outcomes to those achieved by other models that rely just on LSTM or on LSTM that has been optimized.

## 5.2 XAI for explain the forecasting model-based on LIME approach

LIME generates an explanation for a prediction based on the components of an interpretable model that resemble the black-box model near the point of interest.

Fig 16 illustrates the results of the LIME approach to explain specific predictions; it displays the solar DC power production predicting results with each attribute.

LIME provides explainability on a local scale in the form of an explanation for events that take place in close proximity to a prediction. For case "A" in Fig 16, in terms of their numerical

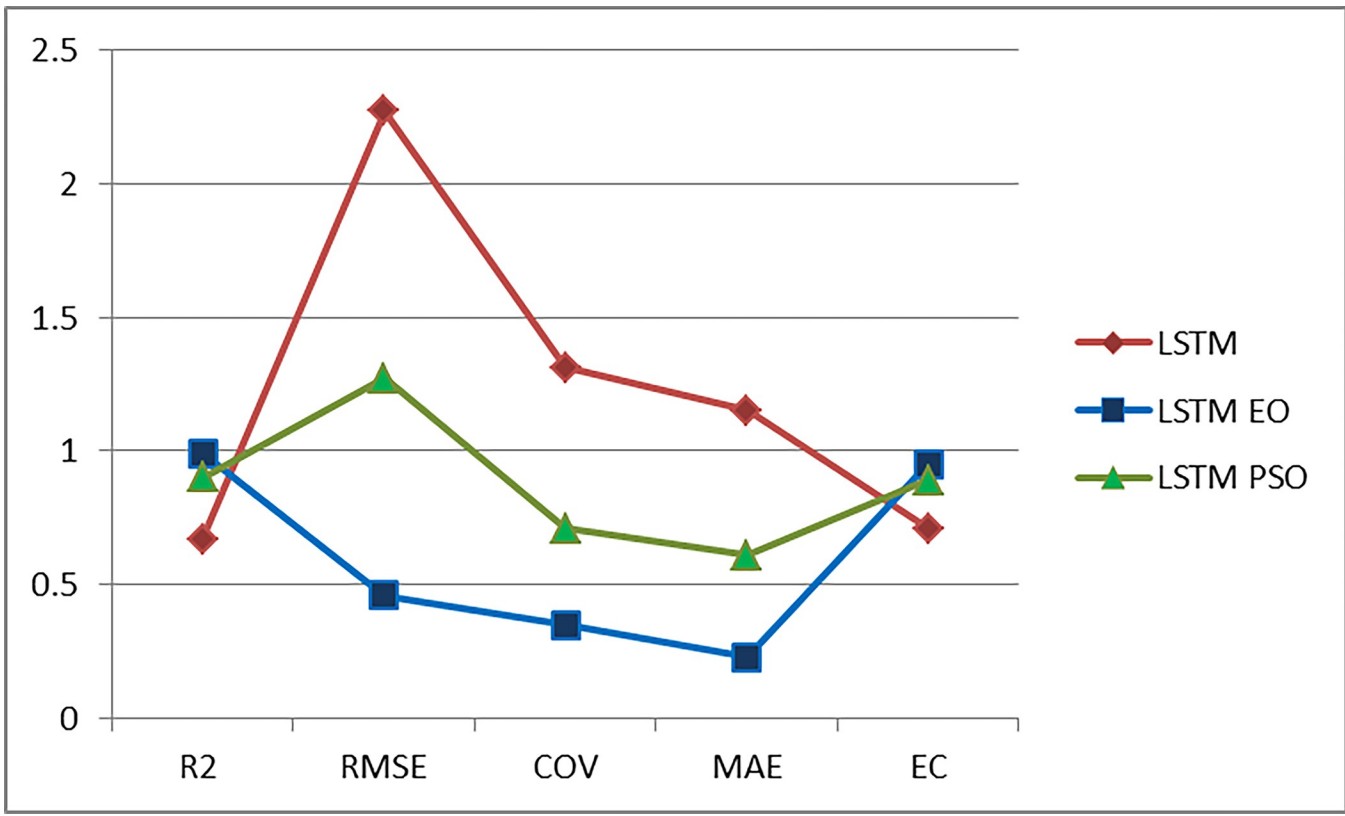

**Fig 15. Experimental models result: LSTM vs LSTM EO vs. LSTM-PSO.**

contribution, the IRRADIATIO**N** is the important feature, whereas the AMBIENT TEMPER-ATURE and MODULE TEMPERATURE are the least important. It is possible to see the impact that each characteristic makes, whether it be good or negative, by looking at the explanations. For instance, AMBIENT TEMPERATURE and MODULE TEMPERATURE have a positive influence, whereas IRRADIATIO**N** has a negative effect on predictions.

For case "D" in Fig 16, in terms of their numerical contribution, the IRRADIATIO**N** and MODULE TEMPERATURE are the important feature, whereas the AMBIENT TEMPERA-TURE is the least important. It is possible to see the impact that each characteristic makes, whether it be good or negative, by looking at the explanations. For instance, AMBIENT TEM-PERATURE and has a positive influence, whereas IRRADIATIO**N** ad MODULE TEMPERA-TURE has a negative effect on predictions and so on in all cases.

As shown before, the XAI permits an explanation of the factors influencing the predicted power output of the solar capacity across various environmental circumstances. However, some researchers employ Maximum Power Point Tracking (MPPT) techniques [52] and electricity distribution burdens [53] that may influence the amount of power generated.

## 6. Conclusion and future works

In this paper the LSTM model is proposed to forecast the power generated by the solar system under different environmental conditions. The performance of LSTM is evaluated in comparison to that of Decision DT and LR. It was demonstrated that the LSTM model performed more effectively than both the DT and LR models when the outcomes were in comparison. For enhancing the results he EO optimizer is proposed for tuning the LSTM hyper parameter.

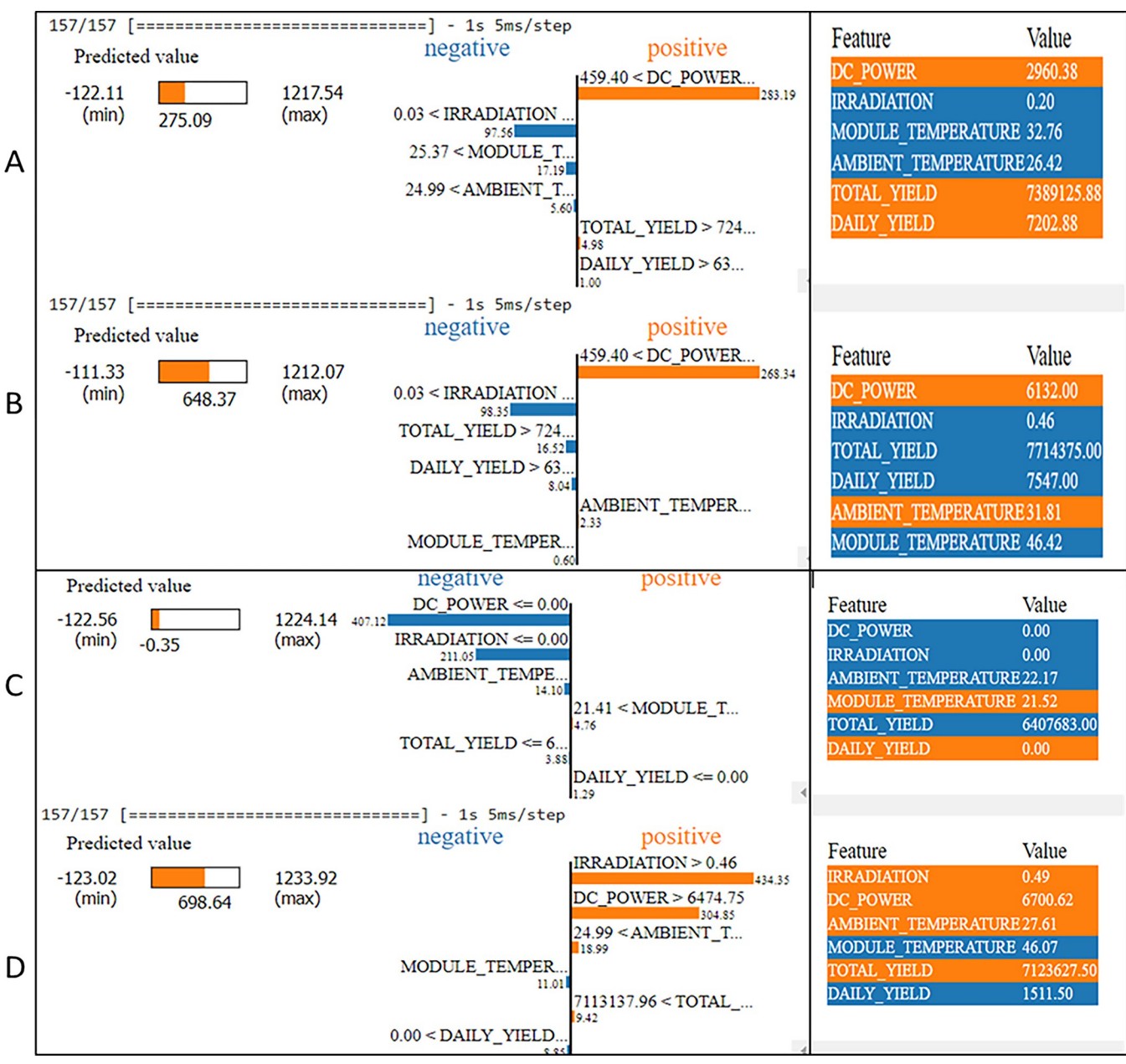

**Fig 16. XAI based LIME algorithm results for solar power generation forecasting model.**

The proposed model produced; $R^2$, RMSE, COV, MAE, and EC values of 0.99, 0.46, 0.35, 0.229, and 0.95, respectively. These results improve the performance before turning hyper-parameter using the EO optimizer, which had $R^2$, RMSE, COV, MAE, and EC values of 0.67, 2.21, 1.31, 1.15, and 0.71 respectively.

Additionally, the XAI-based LIME algorithm was used to explain the results, which helps improve the transparency and interpretability of the model's predictions. This algorithm was successful in identifying the most important features that affected solar power generation, including weather conditions, time of day, and solar panel tilt angle.

In conclusion, the proposed X-LSTM-EO model, along with the use of the XAI-based LIME algorithm, offers a more accurate and transparent method for predicting solar power

generation in solar plant systems. These findings have important implications for developing and deploying renewable energy sources, such as solar power.

The proposed model in this paper exhibits high prediction accuracy under various environmental conditions, demonstrating its universality and accuracy. It reduces uncertainty in PV power generation and safely integrates large-scale PV power generation into micro grids, reducing operating costs and improving efficiency and safety. It also introduces a new PV power generation forecast research direction: clustering data and noise reduction can reduce uncertainty. The research in this paper does not account for harsh weather conditions such as thunderstorms, sand, and dust.

Our future work will incorporate an efficient maximum power point tracking (MPPT) method into our proposed model. This method is crucial for enhancing the efficiency of PV power generation systems. In addition, we will consider the distribution of electrical loads in the proposed model. Finally, we will treat the limitations of our work by considering the harsh weather conditions.

## Author Contributions

**Conceptualization:** Aboul Ella Hassanien.

**Data curation:** Ashraf Darwish.

**Formal analysis:** Ashraf Darwish, Vaclav Snasel.

**Methodology:** Rizk M. Rizk-Allah, Vaclav Snasel.

**Software:** Lobna M. Abouelmagd.

**Supervision:** Aboul Ella Hassanien.

**Validation:** Rizk M. Rizk-Allah, Lobna M. Abouelmagd.

**Visualization:** Lobna M. Abouelmagd.

**Writing – original draft:** Rizk M. Rizk-Allah, Lobna M. Abouelmagd, Ashraf Darwish.

**Writing – review & editing:** Ashraf Darwish, Vaclav Snasel, Aboul Ella Hassanien.

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
