## [Decision Letter · Decision Letter 0]

8 Apr 2024

PONE-D-24-10041Explainable AI and Optimized Solar Power Generation Forecasting Model based on Environmental ConditionsPLOS ONE

Dear Dr. Hassanien,

Thank you for submitting your manuscript to PLOS ONE. After careful consideration, we feel that it has merit but does not fully meet PLOS ONE’s publication criteria as it currently stands. Therefore, we invite you to submit a revised version of the manuscript that addresses the points raised during the review process.

We look forward to receiving your revised manuscript.

Kind regards,

Praveen Kumar Donta, Ph.D.

Academic Editor

PLOS ONE

Journal Requirements:

“no”

Reviewers' comments:

Reviewer's Responses to Questions

**Comments to the Author**

1. Is the manuscript technically sound, and do the data support the conclusions?

Reviewer #1: Yes

Reviewer #2: Partly

Reviewer #3: Yes

2. Has the statistical analysis been performed appropriately and rigorously? 

Reviewer #1: Yes

Reviewer #2: No

Reviewer #3: Yes

3. Have the authors made all data underlying the findings in their manuscript fully available?

Reviewer #1: No

Reviewer #2: No

Reviewer #3: Yes

4. Is the manuscript presented in an intelligible fashion and written in standard English?

Reviewer #1: Yes

Reviewer #2: No

Reviewer #3: Yes

5. Review Comments to the Author

Reviewer #1: The manuscript entitled "Explainable AI and Optimized Solar Power Generation Forecasting Model based on Environmental Conditions”, presents a hybrid machine learning model, X-LSTM-EO and equilibrium optimizer (EO) to predict PV output power with high reliability. In this work, the authors proposed a framework for predicting power generation by using LSTM and component optimization of model's hyper-parameters by using EO component. The proposed work is worthy of investigation. The following are comments on this work to improve the readability of the article.

1)In the introduction, please correct the statement that which increases the power generation, and which decreases the power generation, “Temperature and solar irradiance changes can cause power generation to increase or decrease suddenly.”

2)Please give further details of “original model”, which is mentioned in the abstract.

3)Give a glimpse of software/methodology of the proposed model in the abstract.

4)Keywords are Vogue. Ideally, the keywords should be between 4-6. I suggest the following suitable keywords for this work. “1) Forecasting of solar power generation, ,2) Equilibrium Optimizer, 3) Long short-term memory (LSTM), 4) explainable artificial intelligence (XAI), 5) Local Interpretable and Model-independent Explanation (LIME), 6) Deployment of solar power plants.

5)Please provide the source data and code of the study.

6)A continuity is missing for starting of third paragraph of introduction. So, please merge it with the 2nd paragraph.

7)Introduction section could be improved by maintain a good flow of area of study, major literature and contributions.

8)How does the various MPPT techniques impacts the proposed study? Can the MPPT be investigated in this work?

9)Conclusion is needed to be improved by ignoring the repeated problem statement.

10)In the present study, only the input parameters for generation of power is considered. How could the study go if output loads considered? And further, is this work applied to constant power source generating systems such as small hydro-power systems, if yes, then pls give a future direction by considering the following latest references on constant speed-driven generating stations.

https://doi.org/10.1016/j.renene.2022.06.051

https://doi.org/10.1016/j.rineng.2024.101761

Reviewer #2: The submitted manuscript titled 'Explainable AI and Optimized Solar Power Generation Forecasting Model based on Environmental Conditions' presents a study of using optimized long short term memory algorithm to predict energy production from the sun using solar irradiance based on data from India available on Kaggle. However, the link provided at [36] did not lead to the dataset in question. The authors are advised to review all references, and update review of literature with recent publications from 2022-2024 regarding short-and long-term solar irradiance forecasting. While also subjecting the data to greater scrutiny.

The novelty of the work as described in the manuscript is insufficient to merit further consideration. The authors must improve the state of the art of solar energy production forecasting using LSTM by a significant margin, and better highlight their contributions in the manuscript. The figures also need considerable effort to improve quality and readability, please avoid using screen shots or screen grabs and rely on professional graphics.

The using of english language should also be reviewed. There are some instances where the meaning is not properly conveyed to the reader. While it doesn't pose a challenge to native speakers, it would be beneficial for the manuscript to undergo a thorough review of academic english language usage.

Reviewer #3: The study developed a LSTM based model to forecast the solar power using explainable AI. The study is interesting and can be consider after strictly revision of following points.

1) Reduce the length of Abstract

2) As the LSTM used in the work, the literature of LSTM must be focused at maximum. study based on contrast models like GRU and BILSTM should also discussed in the literature. For reference, following studies should consider:

https://www.tandfonline.com/doi/abs/10.1080/15435075.2022.2143272

https://link.springer.com/article/10.1007/s12145-023-01020-9

https://ajse.aiub.edu/index.php/ajse/article/view/212

3) Describe novelty and reason of development of such model only.

4) Section-1 have the literature & contributions based on studies. Why separate section -2 (related work) ?

5) Point wise conclusion can be provided.

6) Limitation of Work OR challenges ??

7) Provide papers of last 5 years in literature. some are from year 2015,2017. Refer comment 2

8) Discussion must consider the topologies and complexities of the models.

9) Same for Comparision with different models.

6. PLOS authors have the option to publish the peer review history of their article (what does this mean?). If published, this will include your full peer review and any attached files.

Reviewer #1: **Yes: **V. B Murali Krishna

Reviewer #2: No

Reviewer #3: **Yes: **Dr. Pardeep Singla

---

## [Author Response · Author response to Decision Letter 0]

21 Apr 2024

Reviewer # 1

1)In the introduction, please correct the statement that which increases the power generation, and which decreases the power generation, “Temperature and solar irradiance changes can cause power generation to increase or decrease suddenly.”

Response 

 Thank you for your comment, it is revised and updated Page 2-5 

2)Please give further details of “original model”, which is mentioned in the abstract.

Response 

Thank you. The clarification was added to the abstract

" These results improve the performance of the original model that acts without hyper-parameter optimization " Page1 

3)Give a glimpse of software/methodology of the proposed model in the abstract.

Response 

 The clarification was added to the abstract.

The proposed model is implemented utilizing Tensorflow and Keras within the Google Colab environment. 

 Page1

4)Keywords are Vogue. Ideally, the keywords should be between 4-6. I suggest the following suitable keywords for this work. “1) Forecasting of solar power generation, ,2) Equilibrium Optimizer, 3) Long short-term memory (LSTM), 4) explainable artificial intelligence (XAI), 5) Local Interpretable and Model-independent Explanation (LIME), 6) Deployment of solar power plants.

Response 

 Thank you for your recommendation, it is revised and updated Page 2

5) Please provide the source data and code of the study.

Response 

 The source data is cited on page 10 via the hyperlink in reference number 36. Page 10 

6)A continuity is missing for starting of third paragraph of introduction. So, please merge it with the 2nd paragraph.

Response 

 Thank you for your comment, it is revised and the introduction has been updated Page 2-5

7)Introduction section could be improved by maintain a good flow of area of study, major literature and contributions. 

Response 

Thank you for your comment, it is revised and the introduction has been updated Page 2-5

8)How does the various MPPT techniques impacts the proposed study? Can the MPPT be investigated in this work?

Response 

The present study doesn't consider the MPPT techniques, it may be considered in the future work Page 32

9)Conclusion is needed to be improved by ignoring the repeated problem statement.

Response 

Thank you for your comment, it is revised and the conclusion has been updated Page 31

10)In the present study, only the input parameters for generation of power is considered. How could the study go if output loads considered? And further, is this work applied to constant power source generating systems such as small hydro-power systems, if yes, then pls give a future direction by considering the following latest references on constant speed-driven generating stations.

https://doi.org/10.1016/j.renene.2022.06.051

https://doi.org/10.1016/j.rineng.2024.101761

Response 

The proposed Work without electrical loads, and in the case of electrical loads, we will take the load distribution and apply the proposed model to it. Page 32

Reviewer # 2

he submitted manuscript titled 'Explainable AI and Optimized Solar Power Generation Forecasting Model based on Environmental Conditions' presents a study of using optimized long short term memory algorithm to predict energy production from the sun using solar irradiance based on data from India available on Kaggle. However, the link provided at [36] did not lead to dataset in question.

Response 

 It works, this is the screen shot of the link site 

The authors are advised to review all references, and update review of literature with recent publications from 2022-2024 regarding short-and long-term solar irradiance forecasting. While also subjecting the data to greater scrutiny.

Response 

Thank you for your comment, it is revised and the references has been updated Pages 32- 35

The novelty of the work as described in the manuscript is insufficient to merit further consideration. The authors must improve the state of the art of solar energy production forecasting using LSTM by a significant margin, and better highlight their contributions in the manuscript. 

Response 

 thank you for your comment, but our novelty and the contributions in the paper are:

• Deep learning models might not be as accurate because they use traditional optimization methods to find the best internal parameters. These techniques can get stuck in local minima, which leads to finding parameters that aren't as good as they could be. So, this paper solves these problems by adding the LSTM and the equilibrium optimization (EO) to the suggested model. This model is then used to show correctly how solar output power is related to external factors.

• Applying the Equilibrium Optimizer (EO) algorithm for tuning the hyper-parameters the LSTM to enhance the performance of the forecasting, the performance was evaluated in this paper after the application of EO with LSTM.

• Applying PSO optimizer for comparing its results with EO optimizer.

• The utilization of LSTM for effective exploration of the search space without being trapped in local optima areas.

• To understand the forecasting results. XAI's approach called LIME has been applied to explain the performance of the proposed deep learning model, the XAI explained the most important environmental condition that affects the model's forecasting results.

• The propped X-LSTM-EO model proposes a common, accurate model that predicts well under many environmental scenarios. It mitigates PV power generation unpredictability and safely integrates large-scale PV power generation into micro grids, lowering operational costs and boosting efficiency and safety.

The figures also need considerable effort to improve quality and readability, please avoid using screen shots or screen grabs and rely on professional graphics. 

Response 

Thank you for your comment, it is revised and updated 

The using of english language should also be reviewed. There are some instances where the meaning is not properly conveyed to the reader. While it doesn't pose a challenge to native speakers, it would be beneficial for the manuscript to undergo a thorough review of academic english language usage.

Response 

 Thank you for your comment, it is revised and updated 

Reviewer # 3

The study developed a LSTM based model to forecast the solar power using explainable AI. The study is interesting and can be consider after strictly revision of following points.

1) Reduce the length of Abstract 

Response 

Thank you for your comment, it is revised and the abstract has been updated Page 1

2) As the LSTM used in the work, the literature of LSTM must be focused at maximum. study based on contrast models like GRU and BILSTM should also discussed in the literature. For reference, following studies should consider:

https://www.tandfonline.com/doi/abs/10.1080/15435075.2022.2143272

https://link.springer.com/article/10.1007/s12145-023-01020-9

https://ajse.aiub.edu/index.php/ajse/article/view/212

Response 

 Thank you for your comment, it is revised and the introduction has been updated Page 2-5

3) Describe novelty and reason of development of such model only. 

Response 

Thank you for your comment, but our novelty and the contributions in the paper are:

• Deep learning models might not be as accurate because they use traditional optimization methods to find the best internal parameters. These techniques can get stuck in local minima, which leads to finding parameters that aren't as good as they could be. So, this paper solves these problems by adding the LSTM and the equilibrium optimization (EO) to the suggested model. This model is then used to show correctly how solar output power is related to external factors.

• Applying the Equilibrium Optimizer (EO) algorithm for tuning the hyper-parameters the LSTM to enhance the performance of the forecasting, the performance was evaluated in this paper after the application of EO with LSTM.

• Applying PSO optimizer for comparing its results with EO optimizer.

• The utilization of LSTM for effective exploration of the search space without being trapped in local optima areas.

• To understand the forecasting results. XAI's approach called LIME has been applied to explain the performance of the proposed deep learning model, the XAI explained the most important environmental condition that affects the model's forecasting results.

• The propped X-LSTM-EO model proposes a common, accurate model that predicts well under many environmental scenarios. It mitigates PV power generation unpredictability and safely integrates large-scale PV power generation into micro grids, lowering operational costs and boosting efficiency and safety.

4) Section-1 have the literature & contributions based on studies. Why separate section -2 (related work) ?

Response 

Thank you for your comment, it is revised updated Page 2- 5

5) Point wise conclusion can be provided.

Response 

 Thank you for your comment, it is revised and the conclusion has been updated 

6) Limitation of Work OR challenges?? 

Response 

 Thank you for your valuable comment,

The proposed model in this paper exhibits high prediction accuracy under various environmental conditions, demonstrating its universality and accuracy. It reduces uncertainty in PV power generation and safely integrates large-scale PV power generation into micro grids, reducing operating costs and improving efficiency and safety. It also introduces a new PV power generation forecast research direction: clustering data and noise reduction can reduce uncertainty. 

The research in this paper does not account for harsh weather conditions such as thunderstorms, sand, and dust. 

Response 

This part added to the conclusion section. Page 31

7) Provide papers of last 5 years in literature. some are from year 2015,2017. Refer comment 2 

Response 

Thank you for your comment, it is revised updated 

8) Discussion must consider the topologies and complexities of the models.

Response 

 Thank you for your comment. First, Table 9 displays the calculation of the training time for the three examined models throughout the training process. Which indicates the complexity of each one. Second, the section of the experiment, especially Table 7, page 21, explains the topology of our proposed model-based LSTM. Furthermore, the other techniques—linear regression and decision trees—do not resemble LSTM in their topology, hence it isn't very sensible to compare the topologies. Table 9 page 23

9) Same for Comparison with different models. 

Response 

Thank you for your comment, the Comparison exists in section 5.1 , table 13 Page 28

---

## [Decision Letter · Decision Letter 1]

30 Apr 2024

PONE-D-24-10041R1Explainable AI and Optimized Solar Power Generation Forecasting Model based on Environmental ConditionsPLOS ONE

Dear Dr. Hassanien,

Thank you for submitting your manuscript to PLOS ONE. After careful consideration, we feel that it has merit but does not fully meet PLOS ONE’s publication criteria as it currently stands. Therefore, we invite you to submit a revised version of the manuscript that addresses the points raised during the review process.

We look forward to receiving your revised manuscript.

Kind regards,

Praveen Kumar Donta, Ph.D.

Academic Editor

PLOS ONE

Journal Requirements:

Reviewers' comments:

Reviewer's Responses to Questions

**Comments to the Author**

1. If the authors have adequately addressed your comments raised in a previous round of review and you feel that this manuscript is now acceptable for publication, you may indicate that here to bypass the “Comments to the Author” section, enter your conflict of interest statement in the “Confidential to Editor” section, and submit your "Accept" recommendation.

Reviewer #1: All comments have been addressed

Reviewer #3: All comments have been addressed

2. Is the manuscript technically sound, and do the data support the conclusions?

Reviewer #1: Yes

Reviewer #3: Yes

3. Has the statistical analysis been performed appropriately and rigorously? 

Reviewer #1: Yes

Reviewer #3: Yes

4. Have the authors made all data underlying the findings in their manuscript fully available?

Reviewer #1: (No Response)

Reviewer #3: Yes

5. Is the manuscript presented in an intelligible fashion and written in standard English?

Reviewer #1: (No Response)

Reviewer #3: Yes

6. Review Comments to the Author

Reviewer #1: I do appreciate the authors for a good revision.

The following minor comments should be addressed for the final acceptance from my end.

1) Since the authors mention the MPPT is a future work, then they should give at least few reference for it. I suggest authors refer the dedicated review journals and cite appropriate publications.

https://www.sciencedirect.com/journal/renewable-and-sustainable-energy-reviews

2) My earlier comment (comment 10) is partially addressed in the conclusion without proper citations in the conclusion for "distribution of electrical loads".

3) Should give the page numbers for the revised manuscript and line number as you mention in the Comments & Response sheet.

4) Please check the numbering for section and subsections properly and also provide the numbering for every subheadings. Wrong numbering is given for sections 4 and 5.

5) Provide a table for acronyms.

Reviewer #3: Authors Incorporated all comments.

Therefore, paper is recommended for publication.

7. PLOS authors have the option to publish the peer review history of their article (what does this mean?). If published, this will include your full peer review and any attached files.

Reviewer #1: **Yes: **V. B Murali Krishna

Reviewer #3: **Yes: **Dr. Pardeep Singla

---

## [Author Response · Author response to Decision Letter 1]

6 May 2024

based on Environmental Conditions" Article Number: PONE-D-24-10041R1

Reviewer Comment Response Page Number

Reviewer # 3

1) Since the authors mention the MPPT is a future work, then they should give at least few reference for it. I suggest authors refer the dedicated review journals and cite appropriate publications.

https://www.sciencedirect.com/journal/renewable-and-sustainable-energy-reviews

Thank you for your comment, it is revised. the text is updated and reference 49 is added Page 30. 

Line 3:6

2) My earlier comment (comment 10) is partially addressed in the conclusion without proper citations in the conclusion for "distribution of electrical loads".

 Thank you for your comment, it is revised. the text is updated and reference 50 is added Page 30

Lines 3:6

3) Should give the page numbers for the revised manuscript and line number as you mention in the Comments & Response sheet.

 Take into consideration 

4) Please check the numbering for section and subsections properly and also provide the numbering for every subheadings. Wrong numbering is given for sections 4 and 5.

 Thank you for your comment, it is revised and updated 

5) Provide a table for acronyms. List of Acronyms Table is added Page 2

---

## [Decision Letter · Decision Letter 2]

10 Jun 2024

PONE-D-24-10041R2Explainable AI and Optimized Solar Power Generation Forecasting Model based on Environmental ConditionsPLOS ONE

Dear Dr. Hassanien,

Thank you for submitting your manuscript to PLOS ONE. After careful consideration, we feel that it has merit but does not fully meet PLOS ONE’s publication criteria as it currently stands. Therefore, we invite you to submit a revised version of the manuscript that addresses the points raised during the review process.

We look forward to receiving your revised manuscript.

Kind regards,

Praveen Kumar Donta, Ph.D.

Academic Editor

PLOS ONE

Journal Requirements:

Reviewers' comments:

Reviewer's Responses to Questions

**Comments to the Author**

1. If the authors have adequately addressed your comments raised in a previous round of review and you feel that this manuscript is now acceptable for publication, you may indicate that here to bypass the “Comments to the Author” section, enter your conflict of interest statement in the “Confidential to Editor” section, and submit your "Accept" recommendation.

Reviewer #1: All comments have been addressed

Reviewer #4: (No Response)

2. Is the manuscript technically sound, and do the data support the conclusions?

Reviewer #1: Yes

Reviewer #4: Partly

3. Has the statistical analysis been performed appropriately and rigorously? 

Reviewer #1: Yes

Reviewer #4: Yes

4. Have the authors made all data underlying the findings in their manuscript fully available?

Reviewer #1: Yes

Reviewer #4: No

5. Is the manuscript presented in an intelligible fashion and written in standard English?

Reviewer #1: Yes

Reviewer #4: Yes

6. Review Comments to the Author

Reviewer #1: (No Response)

Reviewer #4: 1-The methodology section, particularly the description of the Equilibrium Optimizer (EO) algorithm and its implementation lacks clarity and detailed steps. The current narrative could benefit from a more structured presentation to facilitate comprehension. Including pseudocode or a flowchart would help in understanding the sequential steps of the algorithm. By visualizing the process, readers can better grasp the logic and flow of the EO algorithm.

2-More literature review is need for comparison. Add more studies and make comparative analysis with recent works, here are some examples:

a.Bukhari SM, Moosavi SK, Zafar MH, Mansoor M, Mohyuddin H, Ullah SS, Alroobaea R, Sanfilippo F. Federated transfer learning with orchard-optimized Conv-SGRU: A novel approach to secure and accurate photovoltaic power forecasting. Renewable Energy Focus. 2024 Mar 1;48:100520.

b.Abou Houran M, Bukhari SM, Zafar MH, Mansoor M, Chen W. COA-CNN-LSTM: Coati optimization algorithm-based hybrid deep learning model for PV/wind power forecasting in smart grid applications. Applied Energy. 2023 Nov 1;349:121638.

c.Khan UA, Khan NM, Zafar MH. Resource efficient PV power forecasting: Transductive transfer learning based hybrid deep learning model for smart grid in Industry 5.0. Energy Conversion and Management: X. 2023 Oct 1;20:100486.

3-The data preparation phase mentions combining two datasets, but it lacks specifics on how missing values, outliers, and discrepancies between the datasets were handled. Effective data preprocessing is crucial for the reliability of the model, and a detailed account of these steps is essential. Providing detailed steps on data cleaning, handling missing values, and addressing any discrepancies between the two datasets before integration would ensure transparency and reproducibility of the results.

4-The explanation of evaluation metrics like R2, RMSE, MAE, COV, and EC lacks context and examples of how these metrics are calculated and interpreted. These metrics are pivotal in assessing the model’s performance, and a clear understanding of their computation and significance is necessary. Including a brief explanation with examples for each evaluation metric to illustrate how they are calculated and what they signify in the context of the model’s performance would enhance the manuscript's clarity and educational value.

5-While the paper compares the proposed model with Decision Tree and Linear Regression, it would benefit from a comprehensive comparative analysis with other advanced models, such as Gradient Boosting or other deep learning architectures. This expanded comparative analysis would provide a broader perspective on the proposed model’s performance and demonstrate its robustness and competitiveness against a wider array of contemporary methods.

6-Ensuring consistent use of terminology throughout the manuscript is crucial for clarity. For example, the term "Equilibrium Optimizer" should be uniformly referred to as EO throughout the text.

7-The manuscript contains typos and grammatical errors, such as "the" instead of "The". So, try to improve the English.

8-Some figures and tables are referenced in the text without proper context or description, making it difficult for readers to understand their relevance.

7. PLOS authors have the option to publish the peer review history of their article (what does this mean?). If published, this will include your full peer review and any attached files.

Reviewer #1: **Yes: **V. B Murali Krishna

Reviewer #4: No

---

## [Author Response · Author response to Decision Letter 2]

12 Jul 2024

Response to Review

Dear Prof. Editor in Chief, PLOS ONE Journal

I would like to thank you very much for the time and effort working on the paper and for the valuable comments and revision which has improved the quality of this paper. 

In the next, I reply to the comments on the paper. All corrections are highlighted in the original text of the paper in yellow color. 

Manuscript Title: “Explainable AI and Optimized Solar Power Generation Forecasting Model based on Environmental Conditions" 

Reviewer # 4

Comment 

1-The methodology section, particularly the description of the Equilibrium Optimizer (EO) algorithm and its implementation lacks clarity and detailed steps. The current narrative could benefit from a more structured presentation to facilitate comprehension. Including pseudocode or a flowchart would help in understanding the sequential steps of the algorithm. By visualizing the process, readers can better grasp the logic and flow of the EO algorithm 

Response 

Thank you for your valuable comment, the algorithm has been revised and updated see Page 11, 12

Comment

More literature review is need for comparison. Add more studies and make comparative analysis with recent works, here are some examples:

a.Bukhari SM, Moosavi SK, Zafar MH, Mansoor M, Mohyuddin H, Ullah SS, Alroobaea R, Sanfilippo F. Federated transfer learning with orchard-optimized Conv-SGRU: A novel approach to secure and accurate photovoltaic power forecasting. Renewable Energy Focus. 2024 Mar 1;48:100520.

b.Abou Houran M, Bukhari SM, Zafar MH, Mansoor M, Chen W. COA-CNN-LSTM: Coati optimization algorithm-based hybrid deep learning model for PV/wind power forecasting in smart grid applications. Applied Energy. 2023 Nov 1;349:121638.

c.Khan UA, Khan NM, Zafar MH. Resource efficient PV power forecasting: Transductive transfer learning based hybrid deep learning model for smart grid in Industry 5.0. Energy Conversion and M--

Response

Thank you for your comment. thee updates are doe as follows:

- the literature review part is revised & updated

- the table 1 is added, it contains "Related works for solar power prediction based on AI tools"

- Reference 34,35 ad 36 are added to the paper Pages: 5, 6 7 and 36

Comment

3-The data preparation phase mentions combining two datasets, but it lacks specifics on how missing values, outliers, and discrepancies between the datasets were handled. Effective data preprocessing is crucial for the reliability of the model, and a detailed account of these steps is essential. Providing detailed steps on data cleaning, handling missing values, and addressing any discrepancies between the two datasets before integration would ensure transparency and reproducibility of the results.

Response

 Thanks for your comment.Before data preparation, we do analysis for the data in section 3 which named " Dataset description and analysis", this step proved the normality of the data and it is clear in table 4 which preset the count , mean, std , min, ad max values of the dataset see Page 12 , 13

Comment

4-The explanation of evaluation metrics like R2, RMSE, MAE, COV, and EC lacks context and examples of how these metrics are calculated and interpreted. These metrics are pivotal in assessing the model’s performance, and a clear understanding of their computation and significance is necessary. Including a brief explanation with examples for each evaluation metric to illustrate how they are calculated and what they signify in the context of the model’s performance would enhance the manuscript's clarity and educational value.

Response

Thank you for comment. Section 4.3 called "Model evaluation" explains in details the evaluation metrics 

 Page 22, 23

Comment

5-While the paper compares the proposed model with Decision Tree and Linear Regression, it would benefit from a comprehensive comparative analysis with other advanced models, such as Gradient Boosting or other deep learning architectures. 

This expanded comparative analysis would provide a broader perspective on the proposed model’s performance and demonstrate its robustness and competitiveness against a wider array of contemporary methods.

Response

 Thank you for your valuable comment which proof the robustness and competitive of our proposed model.

- We apply Gradient Boosting algorithm, the results added to table 9. the results of Gradient Boosting ensure the robustness of our work

- The optimizer algorithm (EO)is used to get the optimal architecture of the deep learning (LSTM) 

Page 25

Comment

6-Ensuring consistent use of terminology throughout the manuscript is crucial for clarity. For example, the term "Equilibrium Optimizer" should be uniformly referred to as EO throughout the text.

Response 

Thank you for your valuable comment, it is revised & updated All the paper

---

## [Decision Letter · Decision Letter 3]

16 Jul 2024

Explainable AI and Optimized Solar Power Generation Forecasting Model based on Environmental Conditions

PONE-D-24-10041R3

Dear Authors,

We’re pleased to inform you that your manuscript has been judged scientifically suitable for publication and will be formally accepted for publication once it meets all outstanding technical requirements.

I would like to see some reference enhancements with the latest XAI work. Please consider this!

Kind regards,

Upaka Rathnayake, PhD

Academic Editor

PLOS ONE

Additional Editor Comments (optional):

Reviewers' comments:

Reviewer's Responses to Questions

**Comments to the Author**

1. If the authors have adequately addressed your comments raised in a previous round of review and you feel that this manuscript is now acceptable for publication, you may indicate that here to bypass the “Comments to the Author” section, enter your conflict of interest statement in the “Confidential to Editor” section, and submit your "Accept" recommendation.

Reviewer #5: All comments have been addressed

2. Is the manuscript technically sound, and do the data support the conclusions?

Reviewer #5: Yes

3. Has the statistical analysis been performed appropriately and rigorously? 

Reviewer #5: Yes

4. Have the authors made all data underlying the findings in their manuscript fully available?

Reviewer #5: No

5. Is the manuscript presented in an intelligible fashion and written in standard English?

Reviewer #5: Yes

6. Review Comments to the Author

Reviewer #5: All the comments from the respective reviewers have been resolved

In the abbreviation, LIME, should be model agnostic.

7. PLOS authors have the option to publish the peer review history of their article (what does this mean?). If published, this will include your full peer review and any attached files.

Reviewer #5: No

---

## [Editor Report · Acceptance letter]

23 Jul 2024

PONE-D-24-10041R3 

PLOS ONE

Dear Dr. Hassanien, 

I'm pleased to inform you that your manuscript has been deemed suitable for publication in PLOS ONE. Congratulations! Your manuscript is now being handed over to our production team.

Kind regards, 

on behalf of

Professor Upaka Rathnayake 

Academic Editor

PLOS ONE